# ScalingAR: Scaling Confidence for Autoregressive Image Generation

**Harold Haodong Chen** [* 1 2]  **Xianfeng Wu** [* 3]  **Wen-Jie Shu** [4]  **Rongjin Guo** [5]  **Disen Lan** [6]
**Harry Yang** [2]  **Ying-Cong Chen** [† 1 2]

## Abstract

Test-time strategies have shown remarkable success in improving large language models, but their application to next-token prediction (NTP) autoregressive (AR) image generation remains largely underexplored. Existing test-time scaling (TTS) methods for visual autoregressive models (VAR) rely on frequent partial decoding and external reward models, which are inefficient and often ineffective for NTP-based image generation due to the inherent instability of intermediate decoding results. To address these limitations, we propose `ScalingAR`, a novel test-time scaling framework tailored for NTP-based AR image generation. `ScalingAR` introduces *token entropy* as a confidence signal and operates at two complementary levels: (*i*) *Profile Level*, integrates intrinsic uncertainty and conditional utilization into a unified confidence state, and (*ii*) *Policy Level*, leverages this state for adaptive trajectory pruning and dynamic guidance scheduling. Without requiring early decoding or auxiliary rewards, `ScalingAR` achieves significant improvements across diverse benchmarks. Experiments show that `ScalingAR` **(I)** improves base models by 12.5% on GenEval and 15.2% on TIIF-Bench, **(II)** reduces visual token consumption by 62.0% while outperforming baselines, and **(III)** enhances robustness, mitigating performance degradation by 26.0% in challenging scenarios. These results establish `ScalingAR` as a robust and efficient test-time scaling solution for autoregressive image generation. Our code: `ScalingAR Repository`.

## 1. Introduction

Large language models (LLMs) (Brown et al., 2020; Radford et al., 2019) have demonstrated the capabilities of next-token prediction (NTP) paradigm. This success has renewed interest in applying autoregressive (AR) architectures beyond text, motivating recent visual generative models that represent images in discrete token spaces (Sun et al., 2024; Tian et al., 2024; Li et al., 2024) as shown in Figure 2 (*b*). Compared to diffusion models, which operate over continuous noise trajectories, token-based AR models promise a more unified modality interface.

As the field evolves, the parameters and training data of foundation models (Wang et al., 2024; Yang et al., 2025) have increasingly grown to levels that are inaccessible for most university researchers. In this context, many studies have started to investigate **post-training** methods. Inspired by recent advancements such as GRPO (Shao et al., 2024), a surge of reinforcement learning research has emerged in both language and visual generation domains (Jiang et al., 2025; Cui et al., 2025). Meanwhile, another research avenue focusing on **test-time scaling** (TTS) has emerged (Lightman et al., 2023; Muennighoff et al., 2025; Zuo et al., 2025), which aims to explore *whether a slight increase in computational expense during inference can achieve performance on par with training-time methods, which typically incur much larger costs*.

While test-time scaling has been extensively researched in language models, analogous progress for autoregressive visual generation remains sparse. Images differ from text in three practical ways that complicate direct transfer: (*i*) **holism:** dropping the last 20% of a text sequence may still leave a syntactically valid answer, whereas truncating an image token stream yields an unusable artifact; (*ii*) **objective ambiguity:** many language scaling setups optimize toward a verifiable final answer (*e.g.*, math reasoning), whereas image generation lacks a single ground-truth target; and (*iii*) **early signal scarcity:** partial image token decodes are visually unstable, making premature selection risky. Moreover, recent work TTS-VAR (Chen et al., 2025b) introduced TTS for the next-scale prediction (NSP) paradigm in visual autoregressive model (VAR) (Tian et al., 2024) by predicting images in a coarse-to-fine manner (Figure 2 (*a*)). This intermediate visibility enables reward models to score during scaling but comes with limitations that require predicting large residual token maps at each scale and frequent decoding makes the process inefficient and less suitable for the NTP paradigm.

---

[*]Equal contribution [1]HKUST(GZ) [2]HKUST [3]UNC-Chapel Hill [4]ZODA [5]CityUHK [6]FDU. Correspondence to: Ying-Cong Chen <yingcongchen@usk.hk>.

*Proceedings of the 43rd International Conference on Machine Learning*, Seoul, South Korea. PMLR 306, 2026. Copyright 2026 by the author(s).

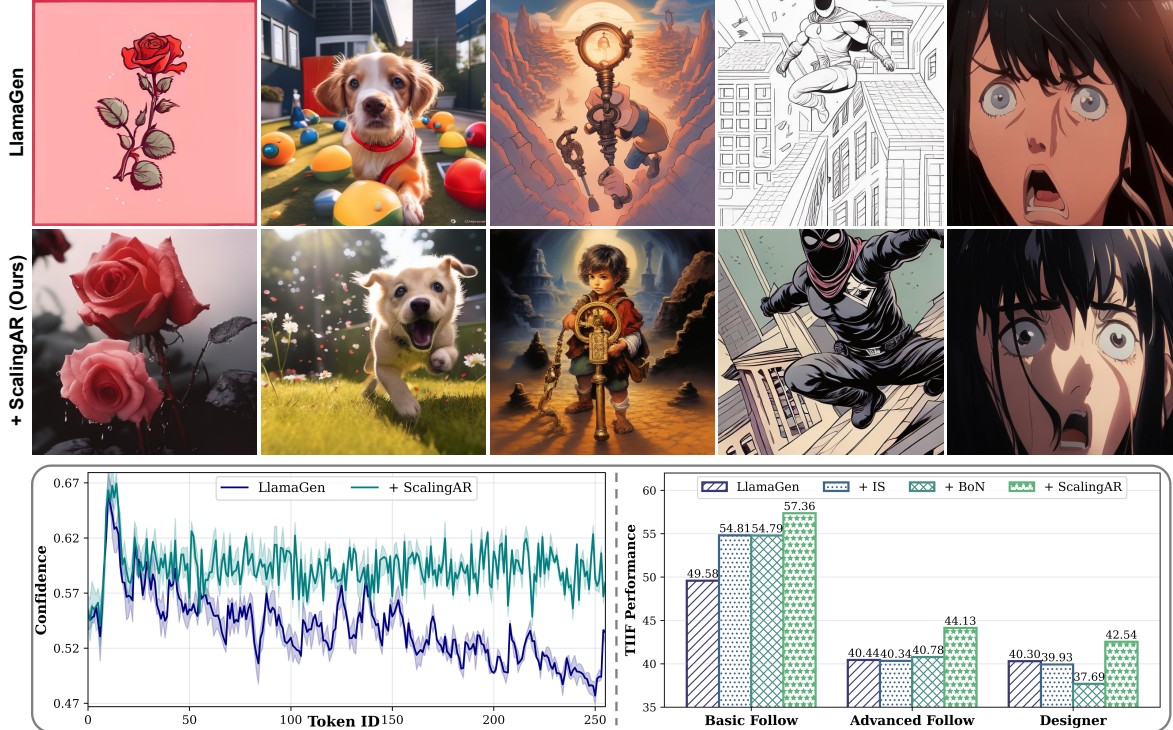

Figure 1. (**Top**) `ScalingAR` significantly improves the quality of autoregressive image generation. Detailed prompts are provided in Appendix §D. (**Bottom Left**) The token confidence trajectory over the generation process. (**Bottom Right**) Performance comparison of `ScalingAR` on TIIF-Bench with classic test-time scaling strategies, *i.e.*, Importance Sampling (IS) and Best-of-N (BoN).

Building on these insights, we introduce `ScalingAR`, a novel test-time scaling framework tailored to the NTP paradigm in autoregressive image generation. Unlike next-scale TTS-VAR, `ScalingAR` eliminates the need for frequent partial decoding and external reward models (as shown in Figure 2 (*d*)), relying solely on intrinsic signals derived from *visual token entropy* and *conditional signals* to profile confidence. Specifically, in response to limitations, `ScalingAR` prunes unreliable trajectories without interrupting generation (*holism*), constructs confidence by combining intrinsic uncertainty and conditional signals (*objective ambiguity*), and extracts stability directly from model probabilities rather than intermediate outputs (*early signal scarcity*). Technically, `ScalingAR` features a two-level design: ♣ *Profile Level*, which constructs a unified confidence state by integrating intrinsic generation stability with conditioning effectiveness; and ♠ *Policy Level*, which leverages this confidence state to prune failing trajectories and dynamically adjust conditioning strength through adaptive termination and guidance scheduling. Our contributions can be summarized as follows:

❶ We propose `ScalingAR`, a novel test-time scaling framework tailored to next-token prediction AR image generation, featuring a two-level design with Profile Level for dual-channel confidence profiling on-the-fly, and Policy Level for trajectory pruning and guidance scheduling.

❷ We for the first time investigate token entropy in visual

token generation. By relying solely on intrinsic signals from the model, `ScalingAR` eliminates the need for frequent early decoding and external reward models, enabling a more efficient and reliable scaling process.

❸ Extensive experiments on both general and compositional benchmarks demonstrate that `ScalingAR` is: **(I) high-performing**, achieving significant performance gains over base models (*i.e.*, LlamaGen and AR-GRPO), by 12.5% on GenEval and 15.2% on TIIF-Bench; **(II) token-efficient**, outperforming classic baselines (*i.e.*, Importance Sampling and Best-of-N) while reducing visual token consumption by 62.0%; and **(III) robust in challenging scenarios**, mitigating performance degradation by 26.0% compared to base models in highly complex generation settings.

## 2. Related Work

**Autoregressive Image Generation.** Autoregressive models have leveraged the scaling capabilities of language models (Yang et al., 2025; Brown et al., 2020; Radford et al., 2019) to generate images. These approaches employ discrete image tokenizers (Van Den Oord et al., 2017; Razavi et al., 2019) in conjunction with transformers, using a next-token prediction strategy. VQ-based methods (Lee et al., 2022; Razavi et al., 2019; Esser et al., 2021), *e.g.*, VQ-VAE (Van Den Oord et al., 2017), convert image patches into index-based tokens, which are then predicted sequentially

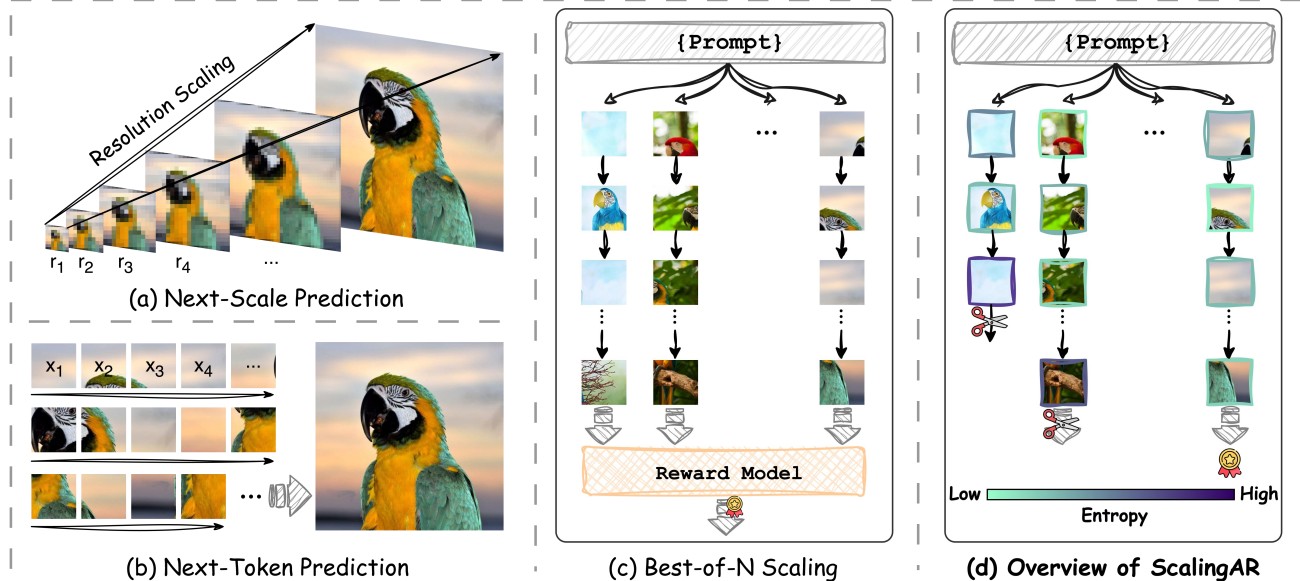

*Figure 2.* (*a*) Next-scale prediction paradigm generates multi-scale token maps coarse-to-fine. (*b*) Next-token prediction paradigm sequentially predicts next image tokens. (*c*) Illustration of Best-of-N sampling that generates multiple candidate and selects the best via voting or scoring. (*d*) Overview of our proposed `ScalingAR`, highlighting its ability to leverage token entropy to early-stop low-confidence samples and identify winning samples without the need for additional reward models.

by a decoder-only transformer. However, these VQ-based AR methods are limited by the lack of scaled-up transformers and the inherent quantization error in VQ-VAE. This has prevented them from achieving performance on par with diffusion models. Recent advancements (Wu et al., 2025a; Yu et al., 2022; Team, 2024) have scaled up AR models for visual generation. Additionally, some variants have been proposed, such as the next-scale prediction paradigm of VAR (Tian et al., 2024; Han et al., 2025), which predicts from coarse to fine token maps, and the parallel token prediction of masked AR (MAR) (Li et al., 2024; Wu et al., 2025b; Fan et al., 2025). Despite these developments, the mainstream approach remains the NTP paradigm, particularly as the field moves towards unified models (Xie et al., 2025; Wang et al., 2024; Ge et al., 2024) that can jointly handle textual and visual tokens. This alignment with language modeling allows for more versatile and scalable architectures.

**Test-Time Scaling.** Current LLMs have increasingly succeeded by allocating substantial reasoning at inference time, a paradigm known as test-time scaling (Snell et al., 2024; Welleck et al., 2024). This scaling can occur along two main axes: (1) Chain-of-Thought (CoT) (Wei et al., 2022) Depth: lengthening a single reasoning trajectory through more thinking steps, often relying on large-scale reinforcement learning with many samples (Yang et al., 2025; Jaech et al., 2024; Guo et al., 2025a) or simpler post-training strategies (Ye et al., 2025; Muennighoff et al., 2025); (2) Parallel Generation: scaling by increasing the number of trajectories and aggregating them, as seen in works like Self-Consistency (Wang et al., 2023) and Best-of-N (Lightman et al., 2023). Recent advances (Kang et al., 2025; Fu

et al., 2025) have also incorporated token entropy for confidence estimation, improving the quality of reasoning traces. However, exploring TTS for AR image generation has been limited. This is due to the holistic nature of image generation, where overall coherence is paramount (see Figure 2 (*c*)), unlike reasoning tasks with well-defined ground truths. Recent work (Guo et al., 2025b) applies Best-of-N to AR image generation by introducing external reward models but without new TTS strategy design itself. Moreover, similar to TTS-VAR (Chen et al., 2025b), it relies on frequent early decoding and rewards. To this end, we propose the first TTS strategy tailored for AR image generation. Notably, we pioneer the exploration of token entropy in image generation, enabling our method to leverage visual token confidence without the need for early decoding or additional rewards.

## 3. Preliminaries

**Next-Token Prediction Autoregressive Modeling.** NTP is a fundamental paradigm in autoregressive models, where the model generates sequences by predicting the next token based on previously generated tokens. The generation process can be mathematically described as follows:

$$p(x_1, x_2, \ldots, x_T) = \prod_{t=1}^{T} p(x_t | x_1, x_2, \ldots, x_{t-1}). \quad (1)$$

This formulation allows the model to leverage past information to inform future predictions, making it particularly effective for sequential data generation.

The training of autoregressive models typically involves maximizing the likelihood of the observed sequences, which

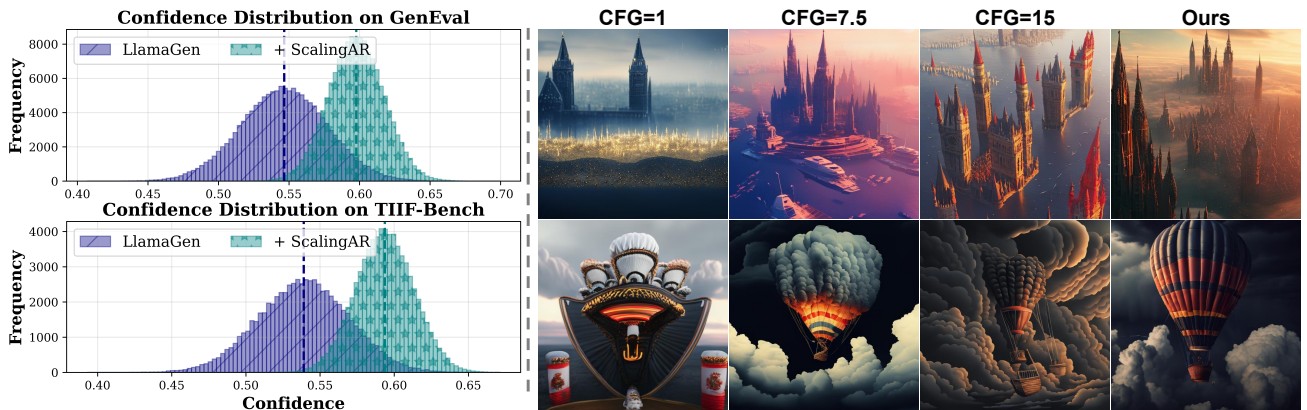

*Figure 3.* (***Left***) Confidence distribution of ScalingAR on GenEval and TIIF-Bench. (***Right***) Illustration of the trade-off between visual quality and semantic alignment with fixed Classifier-Free Guidance (CFG) in AR image generation. *1st: A 35 mm photo of a cityscape resembling Moscow floating in the sky on flying islands. 2nd: The colorful hot air balloon floated near the dark grey storm clouds.*

can be expressed as:

$$L = \sum_{t=1}^{T} \log p(x_t | x_{<t}). \qquad (2)$$

This objective encourages the model to learn the underlying distribution of the data, enabling it to generate coherent and contextually appropriate sequences.

**Token Entropy in Language Modeling.** Token entropy is a critical metric for evaluating the uncertainty associated with the predictions made by language models (Kang et al., 2025). It quantifies the amount of unpredictability in the model's output distribution for a given token. The entropy $H$ at a specific position $i$ in the sequence can be defined as:

$$H_i = - \sum_j p_i(j) \log p_i(j), \qquad (3)$$

where $p_i(j)$ denotes the predicted probability of the $j$-th token in the vocabulary at position $i$. Low entropy indicates high certainty in the prediction, while high entropy reflects greater uncertainty.

Furthermore, token confidence can be derived from the predicted distribution (Fu et al., 2025). The confidence $C_i$ for a token at position $i$ is defined as:

$$C_i = - \frac{1}{k} \sum_{j=1}^{k} \log p_i(j), \qquad (4)$$

where $k$ denotes the number of top tokens considered. High confidence values correlate with sharper distributions, indicating that the model is more certain about its predictions.

## 4. Methodology

To address the challenges of test-time scaling for NTP-based image generation, we propose ScalingAR, a framework that operates on a calibrated confidence state derived solely from the model's internal logits. ScalingAR is designed to operate effectively without relying on frequent partial decoding or external reward models, ensuring com-

putational efficiency while maintaining high-quality generation. The framework consists of two coupled levels: (*i*) **Dual-Channel Confidence Profile** that transforms raw logit statistics into a normalized signal of generation stability (Section §4.1); and (*ii*) **Confidence-Guided Policies** that leverage this state to autonomously prune failing trajectories and modulate guidance strength (Section §4.2).

### 4.1. Dual-Channel Confidence Profile

Autoregressive image generators traditionally treat all partial trajectories as equally promising until completion, as illustrated in Figure 2 (*c*). However, empirical inspection reveals two dominant failure modes during inference that often foreshadow poor final results: ❶ **local instability**, characterized by high entropy pockets and wavering token choices, as shown in Figure 1 (*Bottom Left*) and Figure 3 (*Left*); and ❷ **semantic drift**, where the semantic influence of the prompt gradually fades, resulting in misaligned or aesthetically suboptimal outputs, as shown in Figure 3 (*Right*).

To address these challenges, we introduce the Dual-Channel Confidence Profile, consisting of two complementary channels: ① *intrinsic channel*: captures localized instability and spatial anomalies within the token grid. ② *conditional channel*: quantifies the marginal contribution of textual conditioning to ensure semantic alignment.

#### 4.1.1. INTRINSIC CHANNEL

The first failure mode in visual AR models is local instability, where the model becomes uncertain about the geometric or textural structure of a specific region. This manifests as a high entropy pocket that diffuses into global incoherence. We capture this via two complementary metrics.

**Token-level Confidence.** The raw entropy of the softmax distribution reflects the model's aleatoric uncertainty at each step. However, relying solely on raw entropy is brittle due to varying vocabulary sizes and context-dependent ambi-

guity. Therefore, we formulate a robust token confidence score $s_t^{\text{tok}}$. Let $\pi_t$ denote the softmax distribution over the vocabulary $V$ at step $t$. We combine the normalized entropy $\widehat{H}_t = -\sum_{v \in V} \pi_t(v) \log \pi_t(v)$ with the top-1/top-2 margin $m_t = \pi_t(v_1) - \pi_t(v_2)$, creating a surrogate for decisiveness. Crucially, to filter transient noise inherent in stochastic sampling, we apply an exponential moving average (EMA) to track the confidence trend:

$$s_t^{\text{tok}} = \text{EMA}\left(1 - \text{Norm}(\widehat{H}_t, m_t)\right). \tag{5}$$

A persistent decline in $s_t^{\text{tok}}$ serves as a leading indicator of concept collapse, where the model loses track of the visual structure.

**Worst-block Stability.** A global average of token confidence often masks catastrophic failures occurring in small, localized regions (*e.g.*, a distorted face in a high-quality background). To strictly penalize such local anomalies, we partition the latent grid into non-overlapping blocks and monitor the subset $W_t$ with the highest mean entropy, *i.e.*, the worst $q\%$ of blocks. We derive a stability metric $B_t$ via dynamic normalization:

$$B_t = 1 - \mathcal{N}_{\text{rolling}}\left(\frac{1}{|W_t|} \sum_{k \in W_t} E_k\right). \tag{6}$$

By focusing on the lower bound of spatial stability, $B_t$ ensures that a trajectory is penalized if any critical region fails, aligning with the weakest link principle of visual perception. The final intrinsic score $I_t$ is a calibrated aggregation of the temporal ($s_t^{\text{tok}}$) and spatial ($B_t$) signals.

### 4.1.2. CONDITIONAL CHANNEL

A trajectory may be visually stable (*i.e.*, low entropy) but semantically unrelated to the prompt, *i.e.*, semantic drift. This occurs when the autoregressive prior dominates the conditional probability. To quantify the effective influence of the text prompt $y$, we analyze the information gain provided by the condition at step $t$. Specifically, we compute the KL divergence between the conditional distribution $p(\cdot|x_{<t}, y)$ and the unconditional counterpart $p(\cdot|x_{<t}, \emptyset)$:

$$K_t = \text{KL}\left(p_{c,t} \parallel p_{u,t}\right). \tag{7}$$

Since the magnitude of $K_t$ varies significantly across different prompts and generation stages, absolute thresholding is prone to error. Instead, we employ a dynamic z-score normalization to map $K_t$ into a standardized drift score $\widehat{D}_t \in [0, 1]$. This adaptive normalization renders the metric robust to prompt complexity, requiring no manual tuning per sample. Low values of $\widehat{D}_t$ indicate that the model is ignoring the prompt, signaling a need for intervention.

### 4.1.3. UNIFIED CONFIDENCE STATE

We fuse the intrinsic and conditional signals into a scalar unified confidence score $C_t$. Beyond the instantaneous value,

the dynamics of $C_t$ provide critical information. Failing trajectories often exhibit a "confidence basin", *i.e.*, a prolonged period of low scores. To distinguish between temporary dips and permanent collapse, we track the relative rebound $R_t$:

$$R_t = \frac{C_t - C_{\min}(t)}{|C_{\min}(t)| + \varepsilon}, \tag{8}$$

where $C_{\min}(t)$ is the running minimum. This unified state $S_t = \{C_t, R_t, I_t, \widehat{D}_t\}$ provides a holistic view of the generation health, serving as the basis for our control policies.

### 4.2. Confidence-Guided Policies

With a calibrated state $S_t$, we transition from passive observation to *active test-time control*, enabling dynamic intervention in autoregressive generation. To achieve this, we introduce two lightweight yet effective policies: ① an *adaptive termination* that prunes unpromising trajectories to reclaim computation; and ② a *guidance scheduler* that dynamically modulates CFG scale to balance semantic alignment.

### 4.2.1. ADAPTIVE TERMINATION

A core challenge in test-time scaling is determining *when* to abandon a computation. Static thresholds are suboptimal: a hard prompt may result in naturally lower confidence scores across all valid trajectories, while an easy prompt yields higher scores. A fixed cutoff would thus aggressively prune valid samples in difficult cases. To address this, we propose a distribution-aware pruning mechanism.

**Quantile-Based Thresholding.** We define a dynamic threshold $\theta_\downarrow$ based on the statistics of the current batch. Initialized as the **p**-quantile of the confidence scores $C_t$ across active beams, $\theta_\downarrow$ adapts to the difficulty of the prompt. As generation proceeds, $\theta_\downarrow$ is updated via EMA, ensuring it tracks the non-stationary distribution of token confidence:

$$\theta_\downarrow \leftarrow (1 - \lambda)\theta_\downarrow + \lambda \cdot \text{Quantile}_{\mathbf{p}}(\{C_t\}_{\text{current}}). \tag{9}$$

This ensures that we always prune the relatively worst-performing trajectories, regardless of the absolute confidence scale.

**Recovery Safeguard.** To prevent the premature termination of trajectories experiencing transient instability, we implement a recovery hysteresis. A candidate flagged for pruning ($C_t < \theta_\downarrow$) is granted a probation window. It is only terminated if it fails to demonstrate a significant rebound (*i.e.*, low $R_t$) within this window. This mechanism effectively filters out false positives, preserving diversity while eliminating catastrophic failures.

### 4.2.2. GUIDANCE SCHEDULER

Classifier-free guidance (CFG) creates a trade-off: high guidance improves alignment but reduces diversity and visual quality, while low guidance risks semantic drift. Stan-

*Table 1.* Evaluation on GenEval (Ghosh et al., 2023) and TIIF-Bench (Wei et al., 2025) benchmarks. "Diff.+AR" refers to the unified architecture, and "MAR" indicates the masked AR architecture (Li et al., 2024). We **bold** the best results.

| Method | #Params | Arch. | GenEval | | | | TIIF-Bench | | | |
|---|---|---|---|---|---|---|---|---|---|---|
| | | | Two Obj.↑ | Posit.↑ | Color Attr.↑ | Over.↑ | Basic↑ | Advanced↑ | Designer↑ | Over.↑ |
| DALLE·3 (Betker et al., 2023) | - | Diff. | - | - | - | 0.67 | 78.40 | 68.45 | 62.69 | 72.94 |
| Show-o (Xie et al., 2025) | 1.3B | Diff.+AR | 0.80 | 0.31 | 0.50 | 0.68 | 71.30 | 59.89 | 68.66 | 59.24 |
| LightGen (Wu et al., 2025b) | 0.8B | MAR | 0.65 | 0.22 | 0.43 | 0.62 | 53.99 | 45.76 | 59.70 | 46.42 |
| Infinity (Han et al., 2025) | 2B | VAR | 0.85 | 0.49 | 0.57 | 0.73 | 71.63 | 57.81 | 61.19 | 59.66 |
| Emu3 (Han et al., 2025) | 8.5B | AR | 0.81 | 0.49 | 0.45 | 0.66 | - | - | - | - |
| Janus (Wu et al., 2025a) | 1.5B | AR | 0.68 | 0.46 | 0.42 | 0.61 | - | - | - | - |
| AR-GRPO (Yuan et al., 2025) | 0.8B | AR | 0.27 | 0.02 | 0.03 | 0.31 | 19.59 | 14.91 | 17.91 | 16.22 |
| + IS | 0.8B | AR | 0.47 | 0.08 | 0.07 | 0.44 | 26.00 | 19.03 | 17.62 | 19.84 |
| + BoN | 0.8B | AR | 0.46 | 0.08 | 0.06 | 0.44 | 25.67 | 19.91 | 20.69 | 21.08 |
| + ScalingAR (Ours) | 0.8B | AR | **0.54** | **0.24** | **0.15** | **0.49** | **29.71** | **26.43** | **25.90** | **26.35** |
| LlamaGen (Sun et al., 2024) | 0.8B | AR | 0.34 | 0.21 | 0.04 | 0.32 | 49.58 | 40.44 | 40.30 | 40.35 |
| + IS | 0.8B | AR | 0.21 | 0.11 | 0.02 | 0.14 | 54.81 | 40.34 | 39.93 | 42.44 |
| + BoN | 0.8B | AR | 0.27 | 0.11 | 0.02 | 0.15 | 54.79 | 40.78 | 37.69 | 42.02 |
| + ScalingAR (Ours) | 0.8B | AR | **0.40** | **0.28** | **0.12** | **0.36** | **57.36** | **44.13** | **42.54** | **46.47** |

dard methods use a fixed scale, which is suboptimal as the need for guidance varies throughout the generation process. We propose a closed-loop guidance scheduler that modulates the CFG scale $s_t$ in response to the confidence profile.

The scheduler operates on a compensatory principle: it strengthens guidance when the model shows signs of drift or instability, and relaxes it when generation is confident. Formally, the target scale $s_t^{\text{target}}$ is adjusted as:

$$s_t^{\text{target}} \propto \underbrace{(1 - \widehat{D}_t)}_{\text{Drift Correction}} + \underbrace{\text{Var}(I)}_{\text{Volatility Dampening}} - \underbrace{R_t}_{\text{Diversity Reward}} .$$

(10)

This formulation naturally increases $s_t$ when semantic utilization $\widehat{D}_t$ drops or when intrinsic stability $I$ becomes volatile. Conversely, when the trajectory exhibits a strong rebound $R_t$ (*i.e.*, indicating a return to a high-confidence manifold), the scheduler reduces $s_t$ to encourage visual diversity and prevent over-saturation. This dynamic adaptation allows ScalingAR to navigate the Pareto frontier of quality and alignment more effectively than static baselines.

# 5. Experiments

In this section, we conduct extensive experiments to answer the following research questions:

**RQ1:** Does ScalingAR enhance the quality of generated images?

**RQ2:** Does ScalingAR outperform other TTS strategies for both effectiveness and efficiency?

**RQ3:** How sensitive is ScalingAR to its key components?

**RQ4:** Whether ScalingAR holds advantages over other TTS strategies in both scalability and robustness?

## 5.1. Experimental Settings

**Baselines.** We apply ScalingAR to the advanced models: LlamaGen ($512 \times 512$) (Sun et al., 2024) and AR-GRPO ($256 \times 256$) (Yuan et al., 2025). Since no prior work has explored TTS for the NTP image generation, we focus our

comparisons on the following conventional baselines: Importance Sampling (IS) (Owen & Zhou, 2000) and Best-of-N (BoN) (Lightman et al., 2023). We also provide results on SimpleAR (Wang et al., 2025a) and Janus-Pro (Chen et al., 2025a) in Appendix §B.

**Evaluations.** To evaluate the effectiveness of ScalingAR, we adopt GenEval (Ghosh et al., 2023) and TIIF-Bench (Wei et al., 2025) as primary benchmarks for both general and compositional text-to-image generation capabilities. These benchmarks offer a comprehensive evaluation of the model's ability to produce high-quality and semantically consistent images from text prompts.

## 5.2. Performance & Efficiency Comparison

To answer **RQ1** and **RQ2**, we comprehensively compare ScalingAR against two baselines on general and compositional benchmarks in Table 1, alongside qualitative results, user study, and token consumption comparisons shown in Figure 1, 4, and Figure 5. Key observations are summarized as follows: **Obs.❶** *ScalingAR excels in enhancing both general and compositional generation quality.* As illustrated in Table 1, our ScalingAR consistently outperforms baseline methods (*i.e.*, IS and BoN), which achieve minimal or even negative performance gains, across benchmarks targeting distinct aspects of text-to-image generation. Figure 1 (*Top*) and Figure 4 provide qualitative evidence of ScalingAR's capabilities, showcasing visually superior results that excel in aesthetic quality and semantic alignment, *e.g.*, numerical accuracy, color fidelity, and subject clarity. Furthermore, Figure 5 (*Left*) highlights ScalingAR's effectiveness in aligning image generation with human preferences, as validated through user studies. **Obs.❷** *ScalingAR is a token-efficient test-time AR image generation enhancer.* Figure 5 (*Middle*) demonstrates that ScalingAR consistently surpasses other TTS strategies across benchmarks, requiring fewer visual tokens. Unlike BoN, which relies on external reward models and ex-

| LlamaGen | + IS | + ScalingAR (Ours) | LlamaGen | + BoN | + ScalingAR (Ours) |
|---|---|---|---|---|---|

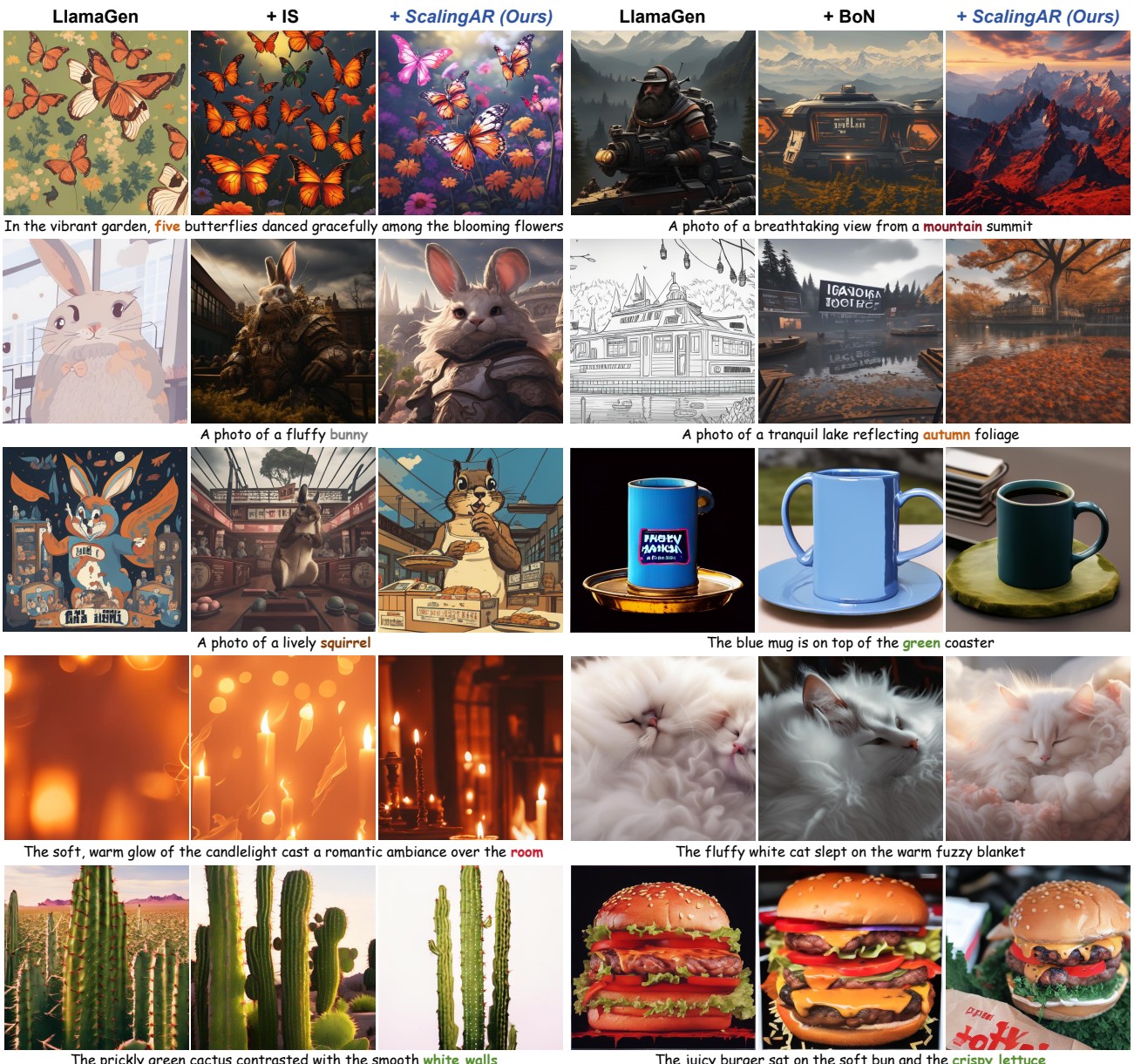

*Figure 4.* Qualitative results of `ScalingAR`. More results on more base models are provided in Appendix §B and §E.

cessive token consumption, `ScalingAR` leverages intrinsic confidence signals to reduce computational overhead while maintaining high-quality outputs.

*Table 2.* Ablation study of `ScalingAR`.

| Method | Bas.↑ | Adv.↑ | Des.↑ | Over.↑ |
|---|---|---|---|---|
| ScalingAR | **57.4** | **44.1** | **42.5** | **46.5** |
| w/o Conditional Channel | 54.1 | 43.1 | 42.2 | 45.1 |
| w/o Worst-Block Stability | 52.3 | 41.8 | 41.4 | 44.2 |
| w/o Token-Level Confidence | 49.6 | 40.4 | 40.3 | 40.4 |

### 5.3. Ablation Analysis

To answer **RQ3**, we perform *step by step* evaluations on TIIF-Bench to analyze the contributions of `ScalingAR`'s

confidence profiles, as detailed in Table 2. We give the following observations: **Obs.❸** *Effectiveness of Intrinsic Signal Profiling.* Removing Token-Level Confidence or Worst-Block Stability both lead to a noticeable drop in performance, highlighting their critical role in capturing fine-grained entropy signals during visual token generation. This demonstrates the effectiveness of intrinsic signal profiling for maintaining local token stability and ensuring high-quality generation. **Obs.❹** *Importance of Condition State Balance.* Table 2 also reveals that removing the Conditional Channel leads to significant degradation. Figure 3 *(Right)* further confirms its critical role in balancing interactions between text guidance and visual generation, ensuring coherent and stable outputs. Since the Policy Level builds

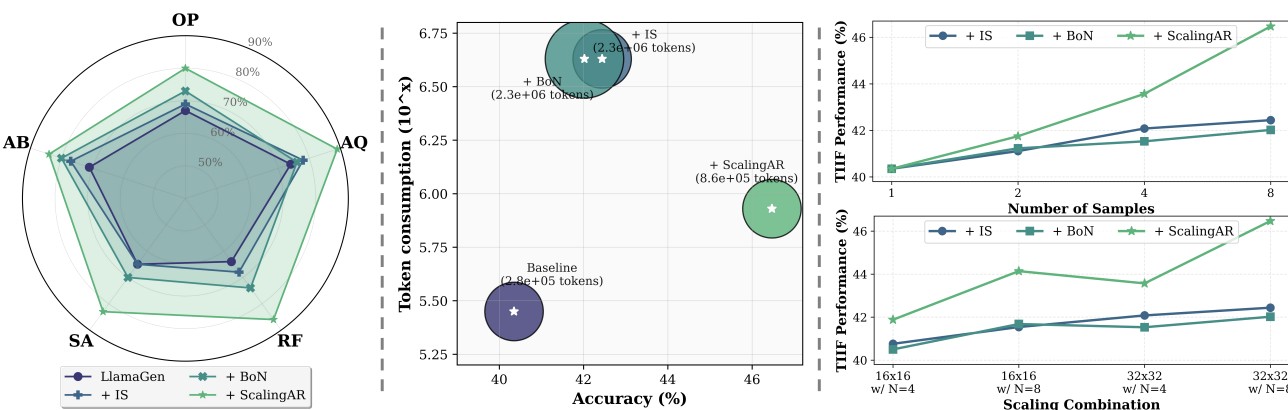

*Figure 5.* (**Left**) User study across five dimensions: overall preference, aesthetic quality, realism fidelity, semantic alignment, attribute binding. (**Middle**) Visual token consumption of `ScalingAR` *vs.* baselines on TIIF-Bench. (**Right**) Scaling width and depth across sample number and token length.

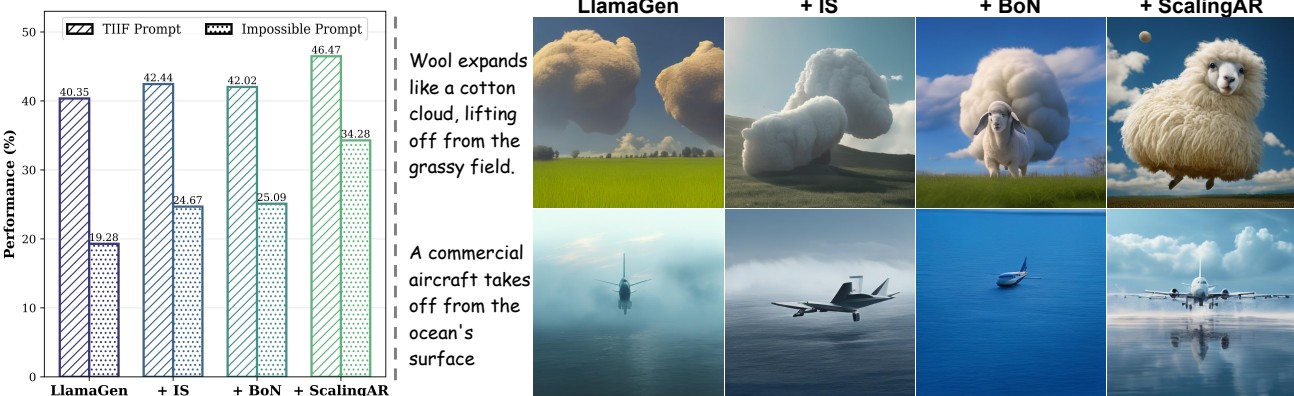

*Figure 6.* Robustness testing with impossible prompt. Detailed prompts are provided in Appendix §D.

upon the Profile Level, we primarily conduct ablation on the Profile Level here and provide Policy Level ablation in Appendix §D with more analysis like wall-clock consumption and FLOPs efficiency.

### 5.4. Scalability & Robustness Analysis

To answer **RQ4**, we compare `ScalingAR` with other TTS strategies (*i.e.*, IS and BoN) in scaling width (*i.e.*, sample number $N$) and depth (*i.e.*, token length), as shown in Figure 5 (*Right*). To further assess the robustness of `ScalingAR`, we adopt the idea of "impossible prompting" (Bai et al., 2025) (*e.g.*, "A young boy ... using chopsticks as a writing instrument, ... in a photo-realistic scene...") to evaluate its performance even when none of the candidates are ideal, with the results presented in Figure 6. Our observations are summarized as follows: **Obs.❺** *ScalingAR unlocks scalable generalization across both width and depth.* As shown in Figure 5 (*Right*), `ScalingAR` consistently outperforms IS and BoN across varying sample numbers and token lengths. This suggests that our scaling strategy enables performance to scale up effectively as scaling width and depth increase, making it a reliable solution for diverse autoregressive tasks. **Obs.❻** *ScalingAR empowers robust generation beyond standard scenarios.* Figure 6 (*Left*)

demonstrates that under impossible prompts for unrealistic scenarios, `ScalingAR` exhibits clear robustness advantages over baselines. Furthermore, Figure 6 (*Right*) confirms that our method achieves more effective scaling when generating under challenging conditions, highlighting its adaptability and reliability in adverse scenarios.

## 6. Conclusion

In this work, we introduce `ScalingAR`, a novel test-time scaling framework tailored to next-token prediction autoregressive image generation. Unlike existing TTS strategies, `ScalingAR` proposes to explore visual token entropy for the first time as intrinsic signals, without relying on partial decoding or external rewards. By adopting a two-level design: Profile Level for calibrated confidence profiling and Policy Level for adaptive pruning and dynamic conditioning, `ScalingAR` achieves phase-aware control, enhancing generation quality with minimal additional token consumption. Comprehensive evaluations on both general and compositional capability benchmarks demonstrate that `ScalingAR` substantially improves the generation quality of existing AR models, along with generalizability and robustness, making it a strong baseline for AR image generation TTS.

## Impact Statement

This paper presents work whose goal is to advance the field of Machine Learning, specifically in efficient autoregressive image generation. A primary societal benefit of our approach is the promotion of sustainable AI practices; by employing distribution-aware pruning to eliminate unpromising generation trajectories, our method significantly reduces computational overhead and energy consumption compared to standard test-time scaling strategies.

On the ethical front, as a inference framework, `ScalingAR` inherits the safety profiles and potential biases of the underlying base models. While `ScalingAR` improves generation fidelity, it does not intrinsically filter harmful content. Therefore, we recommend that real-world deployment of this technology be accompanied by robust safety guardrails and content moderation mechanisms to mitigate risks associated with the generation of misleading or harmful imagery.

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

## A. Further Illustration of Entropy in AR Image Generation

A key motivation behind our `ScalingAR` lies in the observation that high-entropy/low-confidence regions often exhibit greater uncertainty, which increases the likelihood of undesirable outcomes. While high entropy *does not* guarantee poor results, it correlates strongly with elevated error probabilities, making it a critical signal for stabilizing AR image generation.

**Relevant Evidence.** Similar observations have been validated across various domains: ❶ Entropy calibration in language models: Cao et al. (2025) demonstrated that high local token entropy correlates with higher error probabilities, highlighting its role as a risk indicator in generative tasks. ❷ Reinforcement learning mechanisms for reasoning: Works (Cui et al., 2025; Fu et al., 2025; Wang et al., 2025b) for LLM Reasoning treat high-entropy tokens as positions with dense information but unstable decisions or higher error risks. These findings underscore the necessity of carefully managing entropy during generation to balance exploration and stability.

**Connection on `ScalingAR`.** Our method, `ScalingAR`, can be interpreted as a stabilization mechanism that prunes trajectories with low confidence, effectively mitigating the risks associated with high-entropy regions. By focusing on confidence signals, `ScalingAR` ensures that the generation process avoids prolonged instability, leading to improved image quality. In Figure 1 (*Bottom Left*) and Figure 3 (*Left*), we compare the confidence distributions of `ScalingAR` and the base model. The results clearly show that higher token confidence correlates with better image quality, further validating our motivation to leverage confidence signals for trajectory pruning.

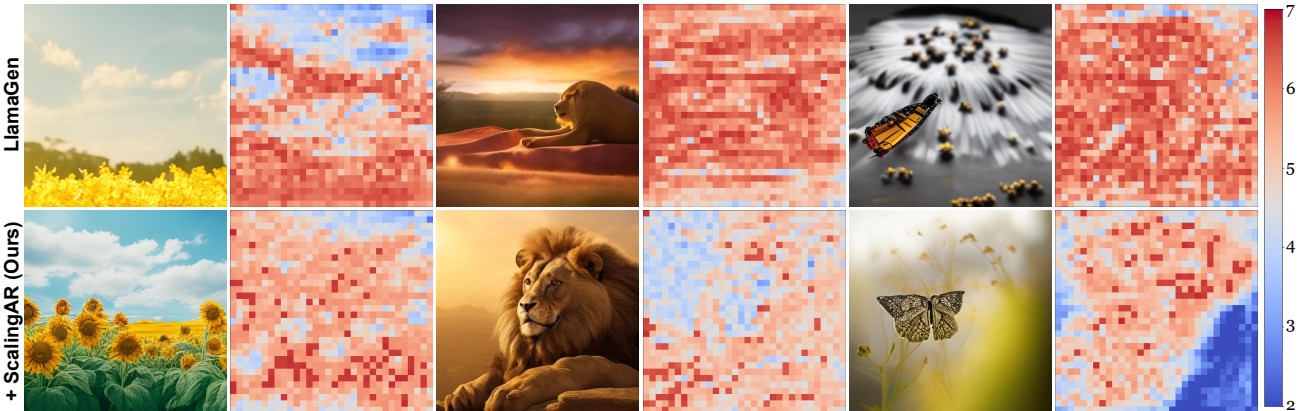

*Figure 7.* Visualization of token entropy. (*1st*) A sunflower field stretching to the horizon under a bright blue sky. (*2nd*) A majestic lion resting on a rocky outcrop in the golden savanna light. (*3rd*) A detailed macro shot of a butterfly on a blooming flower.

**Visualizing Token Entropy in Generated Images.** To provide a more intuitive understanding, we visualize the entropy distributions of generated images in Figure 7. The figure highlights that regions with poor generation quality often correspond to higher entropy, reinforcing the notion that high-entropy tokens are more likely to contribute to undesirable outcomes. `ScalingAR`'s ability to suppress these regions through confidence-based pruning plays a pivotal role in achieving stable and high-quality image generation.

## B. Results of More Base Models

To further validate the generalizability of `ScalingAR`, we deployed our method on two additional AR models: SimpleAR-1.5B (Wang et al., 2025a) and Janus-Pro-1B (Chen et al., 2025a). Importantly, the hyperparameter settings for `ScalingAR` were kept consistent with those used in the main experiments on LlamaGen and AR-GRPO, without any model-specific tuning. This ensures a fair evaluation of `ScalingAR`'s adaptability across different ar-

*Table 3.* Evaluation of `ScalingAR` on more base models on GenEval (Ghosh et al., 2023).

| Method | Two Obj.↑ | Pos.↑ | Color Attri.↑ | Overall↑ |
|---|---|---|---|---|
| SimpleAR | 0.90 | 0.28 | 0.45 | 0.63 |
| + `ScalingAR` (Ours) | **0.93** | **0.36** | **0.51** | **0.67** |
| Janus-Pro | 0.82 | 0.65 | 0.56 | 0.73 |
| + `ScalingAR` (Ours) | **0.87** | **0.69** | **0.61** | **0.77** |

chitectures and scales. Quantitative results in Table 3 and qualitative results in Figure 8 show significant performance improvements for both models, demonstrating `ScalingAR`'s effectiveness and broad applicability as a general-purpose stabilization framework.

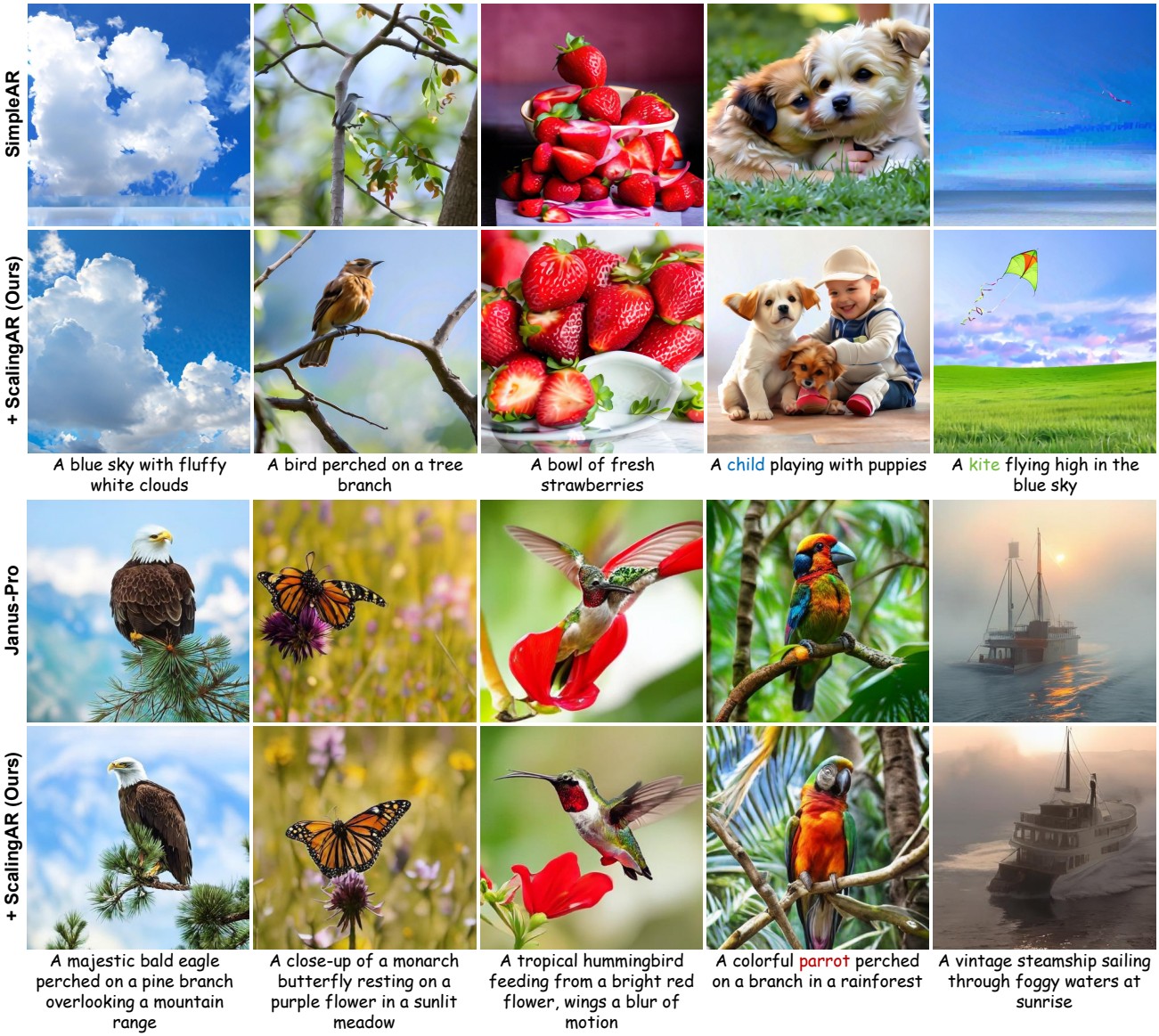

*Figure 8.* Qualitative results of ScalingAR on SimpleAR (***Top***) and Janus-Pro (***Bottom***).

## C. Implementation Details

### C.1. Universal Configuration Strategy

Crucially, ScalingAR utilizes a *single, universal configuration* across all experiments. We do not perform model-specific hyperparameter tuning. The settings detailed below were applied identically to LlamaGen (0.8B), AR-GRPO (0.8B), SimpleAR (1.5B), and Janus-Pro (1B), demonstrating the structural robustness of the proposed method.

### C.2. Hyperparameters

To facilitate reproducibility, we group the parameters by their corresponding stage in the ScalingAR pipeline: (***i***) constructing the confidence state, corresponding to Section §4.1, and (***ii***) executing the guidance policies, corresponding to Section §4.2. The sensitivity analysis of key hyperparameters is placed in Section §D.2.

### C.3. Algorithm Workflow

We conclude the overall algorithm workflow of ScalingAR in Algorithm 1.

| Notation | Parameter Definition | Value | Rationale / Function |
|---|---|---|---|
| *Dual-Channel Confidence Profile* | | | |
| $\alpha_H/\alpha_M$ | Token-Level Weights | 0.5/0.5 | **Robust composite**: combines Entropy (uncertainty) with Margin (decisiveness). |
| $\lambda_{\text{tok}}, \lambda_I$ | Smoothing Factors | 0.2 | Standard EMA for noise filtering in score tracking. |
| $w_{\text{tok}}/w_{\text{blk}}$ | Intrinsic Weights | 0.65/0.35 | Balances token-wise entropy with spatial block stability. |
| $b$ | Spatial Block Size | $4 \times 4$ | Grid resolution for detecting localized failure spots. |
| $q$ | Worst-Case Quantile | 0.1 | Focuses metric on the most unstable regions (weakest link). |
| $w_I/w_D$ | Channel Fusion | 0.75/0.25 | Balances intrinsic stability with conditional alignment. |
| *Confidence-Guided Policies* | | | |
| **p** | Pruning Quantile | 0.2 | **Adaptive threshold**: targets relative bottom-20% of batch. |
| $\lambda_\theta$ | Threshold Update | 0.2 | Update rate for tracking the batch's confidence distribution. |
| $W_0$ | Warm-up Period | 12.5% | Initial generation phase before pruning activation. |
| $\Delta_{\text{rec}}$ | Recovery Window | 32 steps | Hysteresis window to prevent premature termination. |
| $\delta_{\text{rec}}$ | Recovery Threshold | 0.05 | Minimum confidence rebound required to trigger recovery. |
| $\alpha, \beta$ | Guidance Gains | 0.3, 0.4 | Response gain for Drift Correction and Volatility Dampening. |
| $\lambda$ | Diversity Gain | 0.4 | Response gain for Rebound-based guidance relaxation. |

---

**Algorithm 1 ScalingAR Workflow**

---

1: **Input:** Prompt $y$, Model $p_\theta$, Config $\mathcal{C}$ (weights, smoothing, quantiles). **Output:** Image $\hat{x}$
2: Initialize candidates $\mathcal{S}_0 \leftarrow \{\emptyset\}$ and warm-up threshold $\theta_\downarrow$.
3: **for** $t \leftarrow 1$ **to** $T$ **do**
4:    **if** $\mathcal{S}_{t-1} = \emptyset$ **then**
5:       **break**
6:    **end if**
7:    *// Stage I: Signal Extraction (Batch Update)*
8:    Compute logits $\ell_c, \ell_u$ and raw signals $(I_t^{\text{raw}}, K_t)$ for all $s \in \mathcal{S}_{t-1}$
9:    Update State $S_t(s) \leftarrow \{I_t, \widehat{D}_t, C_t, R_t\}$ using EMA and Rolling Stats
10:   *// Stage II: Dynamic Guidance & Sampling*
11:   **for all** $s \in \mathcal{S}_{t-1}$ **do**
12:      Calculate adaptive scale $s_t \leftarrow s_{\text{base}} + \alpha(1 - \widehat{D}_t) + \beta \text{Var}(I) - \gamma R_t$
13:      Sample $x_t \sim \text{softmax}(\ell_u + s_t \cdot (\ell_c - \ell_u))$ and append to $s$
14:   **end for**
15:   *// Stage III: Adaptive Termination*
16:   Update global threshold $\theta_\downarrow \leftarrow \text{EMA}(\theta_\downarrow, \text{Quantile}_{\mathbf{p}}(\{C_t(s)\}))$
17:   Prune $s$ if $C_t(s) < \theta_\downarrow$ **and** $R_t < \delta_{\text{rec}}$ (not recovering)
18:   $\mathcal{S}_t \leftarrow$ survivors after pruning
19: **end for**
20: **return** Decode($\hat{s}$), where $\hat{s} \leftarrow \arg\max_{s \in \mathcal{S}_T} C_T(s)$

---

# D. More Experimental Settings and Analysis

## D.1. More Details of Experimental Settings

**Captions of Figure 1.** For qualitative results in Figure 1 (*Top*), we further detail the prompts here:

- **1st**: *"A red rose in full bloom sits on the top, above a pink rosebud."*

- **2nd**: *"A photo of a cute puppy playing in a sunny backyard."*

- **3rd**: *"A young boy holding a mysterious key, embarking on an adventure through various landscapes to find hidden treasure."*

- **4th**: *"A masked hero jumping from a rooftop, comic book style with bold outlines and dialogue bubbles."*

- **5th**: *"A close-up of an anime woman's face with a shocked expression, featuring dark hair, drawn in the anime style. The image showcases colorful animation stills, close-up intensity, soft lighting, a low-angle camera view, and high detail."*

**Baseline Setup.** We benchmark against Importance Sampling (IS) and Best-of-N (BoN) following the protocols established in TTS-VAR (Chen et al., 2025b). We utilize ImageReward (Xu et al., 2023) as the external preference model due to its

alignment with human aesthetics, and set the primary candidate sample size to $N = 8$. For both baselines, the CFG scales are maintained at the default values of the respective base models to ensure a controlled comparison.

Notably, while BoN is often theoretically regarded as a performance upper bound via exhaustive selection, our method consistently surpasses it. We attribute this to the fundamental difference between *passive selection* and *active rectification*. BoN restricts itself to selecting from static trajectories and is therefore bound by the quality of the external reward proxy. In contrast, our approach employs closed-loop control, dynamically modulating guidance pressure to actively stabilize generation. This allows the model to correct intrinsic drift and access superior regions of the generation manifold that are unreachable by static decoding strategies, regardless of the selection budget.

**Robustness Testing.**   To evaluate the robustness of `ScalingAR`, we further employ prompts from IPV-TXT from Impossible Videos [ICML'25] (Bai et al., 2025). Specifically, we filtered prompts suitable for image generation from IPV-TXT, then employed Impossible Prompt Following (IPF) as the evaluation metric, which measures the alignment between generated images and the semantic intent of impossible prompts. Following (Bai et al., 2025), we employed GPT-4o to perform binary judgments on each image based on prompt adherence. For qualitative results in Figure 6 (*Right*):

- **1st:** *"A sheep peacefully grazing in a realistic meadow suddenly defies gravity as its wool expands dramatically, causing its body to balloon up like a cotton cloud. The fluffy animal then lifts off from the grassy field and drifts upward into the blue sky, its transformed woolly coat acting like a natural balloon."*

- **2nd:** *"A commercial aircraft inexplicably takes off from the ocean's surface as if the water were a solid runway, defying physics in this photo-realistic scene. The calm, glassy sea appears to have transformed into a firm platform, allowing the plane to accelerate and lift off smoothly, with spray trailing behind its wheels like it would on a wet tarmac."*

**User Study.**   We conducted a user study to evaluate human preferences using the mean opinion score (MOS) metric. We designed a user-friendly interface to facilitate the evaluation process and collected feedback from a total of $15$ volunteer participants. The detailed instructions provided to the participants are as follows:

---

**User Study: Autoregressive Image Generation**

Thank you for participating in our user study! Please follow these steps to complete your evaluation:

1. **Image Generation:**  Carefully read the target prompt provided, and then view the provided images.
2. **Scoring Criteria:** Assign a score to each generated image based on the following aspects (1 being the lowest, 5 being the highest):

- *Overall Quality:* The overall perceived quality and appeal of the generated image.

- *Aesthetic Quality:* The visual aesthetics, composition, and artistic merit of the image.

- *Realism Fidelity:* How realistically and faithfully the image captures the intended scene or subject matter.

- *Semantic Alignment:* How well the generated image aligns with and represents the meaning of the textual prompt.

- *Attribute Binding:* The degree to which the image accurately depicts the specific attributes and details described in the text.

3. **Submission:** Click the "Submit Scores" button to submit your scores.

**Notations:**
1. We observe that the edge browser is not fully compatible with our interface. Chrome is recommended.
2. Remember to click the "Submit Scores" button after your evaluation.
3. If you see that images and the score sliders are not aligned, shrinking your page usually works.
4. If the page is not responsive for a long time, please try to refresh it.
5. If you have any questions, please directly ping us. Thank you for your time and effort!

---

### D.2. More Analysis

**Analysis of Ablation on Policy Level.**   While ablation study (Table 2) in main text focuses on the Profile Level, we conducted additional ablation studies to evaluate the contributions of the Policy Level, which builds upon the Profile Level,

as shown in Table 4. (*i*) The "Termination Only" setup improves performance across all metrics, highlighting its ability to prune low-confidence trajectories and mitigate failure modes, ensuring stable generation. (*ii*) The "Scheduler Only" setup also yields notable gains, demonstrating its effectiveness in dynamically modulating

*Table 4.* Ablation of Policy Level.

| Method | Basic↑ | Advanced↑ | Design↑ | Overall↑ |
|---|---|---|---|---|
| LlamaGen | 49.6 | 40.4 | 40.3 | 40.4 |
| + Termination Only | 54.1 | 43.1 | 42.2 | 45.1 |
| + Scheduler Only | 53.6 | 42.0 | 41.0 | 43.8 |
| + ScalingAR (Ours) | **57.4** | **44.1** | **42.5** | **46.5** |

conditioning strength to balance semantic alignment and diversity. (*iii*) Integrating both mechanisms achieves the best results, showing their complementary roles in improving generation quality and efficiency. These results validate the Policy Level as essential for enhancing autoregressive image generation.

**Computational Efficiency Analysis.** We profile the computational overhead of ScalingAR against the base model and the BoN on GenEval, using NVIDIA H200 GPUs. Detailed statistics are reported in Table 5.

*Table 5.* Computation consumption comparison on GenEval with NVIDIA 140G H200 GPU.

| Method | N | Per-step WC (s) | Overall WC (s) | Matched Tokens/Img | FLOPs (TFLOPs) | Memory (GB) | Performance |
|---|---|---|---|---|---|---|---|
| LlamaGen | 1 | 0.024 | 24.93 | 1024 | 5.60 | 6.44 | 0.32 |
| + BoN | 8 | 0.025 | 218.44 | 8192 | 39.12 | 48.72 | 0.15 |
| + ScalingAR (Ours) | 8 | 0.029 | 69.56 | 2350 | 4.23 | 18.16 | 0.36 |

- **Wall Clock Latency:** Standard test-time scaling (BoN, $N = 8$) incurs a prohibitive latency penalty, increasing overall wall clock (WC) time to 218.44s. In contrast, ScalingAR effectively decouples sample width from latency. By actively pruning trajectories, it reduces the overall WC time to 69.56s, achieving a $3.1\times$ speedup over BoN while maintaining superior generation quality.

- **Resource Consumption:** The efficiency gains are further evidenced in compute and memory usage. BoN requires generating $8,192$ tokens per instance, resulting in massive computational demand (39.12 TFLOPs) and high peak memory usage (48.72 GB). Conversely, ScalingAR processes only $2,350$ tokens on average, *i.e.*, a $71\%$ reduction in token consumption compared to BoN. This aggressive pruning translates directly to a dramatically lower compute footprint (4.23 TFLOPs) and reduced memory overhead (18.16 GB), making test-time scaling feasible for resource-constrained environments.

**Analysis of Global Confidence & Guidance Weights.** Figure 9 presents a detailed analysis of the impact of weights of unified confidence and guidance scheduler on the performance of ScalingAR on the TIIF-Bench. ❶ **Unified Confidence** (Figure 9 (*Top*)): Varying the balance between the Intrinsic ($w_I$) and Conditional ($w_D$) channels shows that emphasizing the Intrinsic channel slightly ($w_I/w_D = 0.75/0.25$) achieves the best TIIF-Bench performance across all subsets. This highlights the importance of capturing local uncertainty and stability while maintaining semantic alignment. Omitting the Conditional Channel ($1.00/0.00$) degrades performance, confirming its complementary role. ❷ **Guidance Scheduler** (Figure 9 (*Bottom*)): Adjusting the weights $\alpha$, $\beta$, and $\lambda$, which control conditional utilization, intrinsic volatility, and confidence rebound, respectively, reveals that moderate emphasis on intrinsic volatility and rebound ($\beta, \lambda$) improves performance. The weight $\alpha$ peaks at $0.3$, suggesting overemphasis may reduce diversity. This confirms the need for balanced, dynamic guidance to optimize semantic fidelity and diversity.

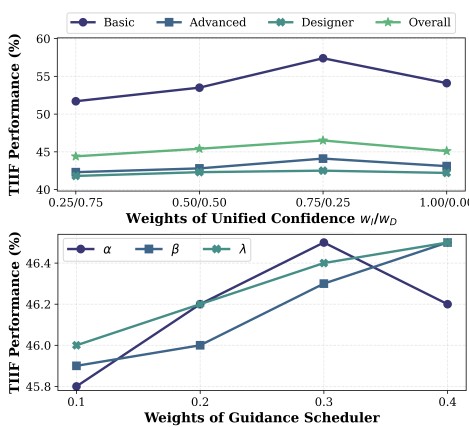

*Figure 9.* Analysis of weights of Unified Confidence (***Top***) and Guidance Scheduler (***Bottom***).

**Analysis of Adaptive Termination Gate.** We further analyze the impact of the confidence threshold quantile **p** and the recovery threshold $\delta_{\text{rec}}$ on the performance and token efficiency of ScalingAR, as illustrated in Figure 10. ❶ **Confidence Threshold** (Figure 10 (*Left*)): The choice of confidence threshold critically balances pruning aggressiveness and generation quality. Setting **p** too low leads to insufficient pruning, resulting in higher token consumption with limited accuracy gains. Conversely, an overly high threshold causes

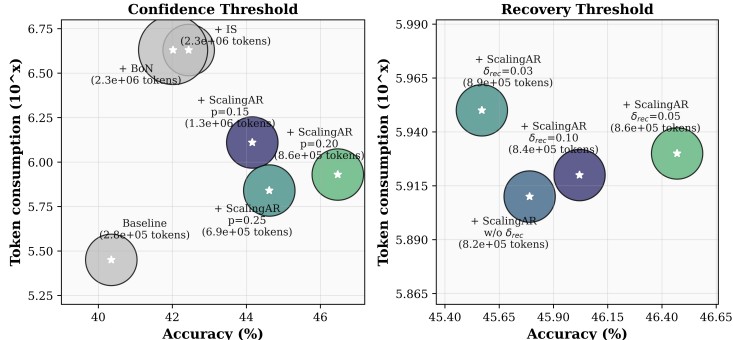

*Figure 10.* Analysis of thresholds of Confidence (***Left***) and Recovery (***Right***).

premature termination of promising trajectories, degrading accuracy despite lower token usage. Our experiments show that an intermediate threshold (*e.g.*, $\mathbf{p} = 0.20$) achieves the best trade-off, significantly improving accuracy while maintaining efficient token consumption compared to both baseline and extreme settings. ❷ **Recovery Threshold** (Figure 10 (*Right*)): The recovery mechanism safeguards against false positives by allowing trajectories to rebound from transient confidence dips. Disabling this mechanism leads to noticeable performance drops, highlighting its necessity. Furthermore, setting the recovery threshold $\delta_{\text{rec}}$ too low or too high adversely affects accuracy and efficiency: a low threshold permits premature recovery of poor trajectories, increasing token cost, while a high threshold delays recovery, risking early termination of viable samples. An optimal value (*e.g.*, $\delta_{\text{rec}} = 0.05$) balances these effects, maximizing accuracy with minimal token overhead.

**Analysis of Local Confidence Weights.** We further analyze the impact of the weighting strategies for Token-Level Confidence $\alpha_H/\alpha_M$ and Worst-Block Stability $w_{\text{tok}}/w_{\text{blk}}$ on the performance of `ScalingAR`, as illustrated in Figure 11. ❶ **Token-Level Confidence Weights** (Figure 11 (*Left*)): Adjusting the balance between entropy-based uncertainty ($\alpha_H$) and margin-based confidence ($\alpha_M$) reveals that prioritizing entropy signals ($\alpha_H/\alpha_M = 0.7/0.3$) achieves the best overall performance across all metrics. This suggests

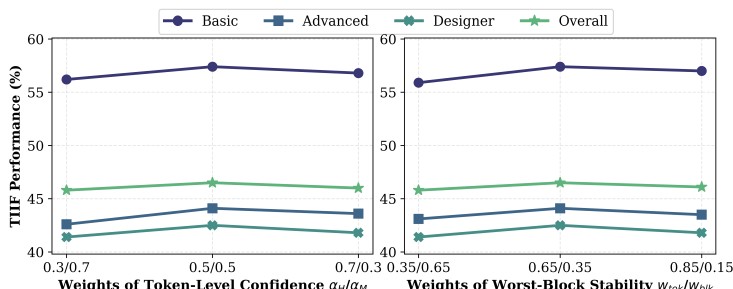

*Figure 11.* Analysis of weights of Token-Level Confidence (***Left***) and Worst-Block Stability (***Right***).

that entropy provides a more robust signal for capturing localized instability during generation. Conversely, overemphasizing margin-based confidence ($\alpha_H/\alpha_M = 0.3/0.7$) leads to performance degradation, particularly in advanced and designer subsets, as it fails to fully capture nuanced instability patterns. A balanced setting ($\alpha_H/\alpha_M = 0.5/0.5$) offers a reasonable trade-off, though slightly underperforms the optimal configuration. ❷ **Worst-Block Stability Weights** (Figure 11 (*Right*)): Varying the balance between token-level confidence ($w_{\text{tok}}$) and block-level stability ($w_{\text{blk}}$) shows that an emphasis on token-level signals ($w_{\text{tok}}/w_{\text{blk}} = 0.85/0.15$) slightly reduces performance, particularly in the advanced and designer subsets, as it underweights spatial anomalies that propagate into global failures. On the other hand, overemphasizing block-level stability ($w_{\text{tok}}/w_{\text{blk}} = 0.35/0.65$) also degrades results, as it may overreact to localized noise. The optimal configuration ($w_{\text{tok}}/w_{\text{blk}} = 0.65/0.35$) balances token-level and block-level signals effectively, achieving the highest scores across most metrics.

# E. Exhibition Board

We provide more comparison results here in Figure 12 on AR-GRPO and Figure 13 on LlamaGen.

# F. Limitation and Future Works

`ScalingAR` pioneers test-time scaling for autoregressive image generation but faces key challenges. AR image modeling involves complex dependencies, making confidence estimation difficult; our exploration of token entropy is a first step but may not fully capture uncertainty and semantic alignment. Additionally, the approach relies on model calibration and entropy signals, which can vary with training and architecture. Future work includes developing finer-grained confidence measures for more precise scaling, and integrating entropy-based signals into both training-time and test-time to create a more unified pipeline.

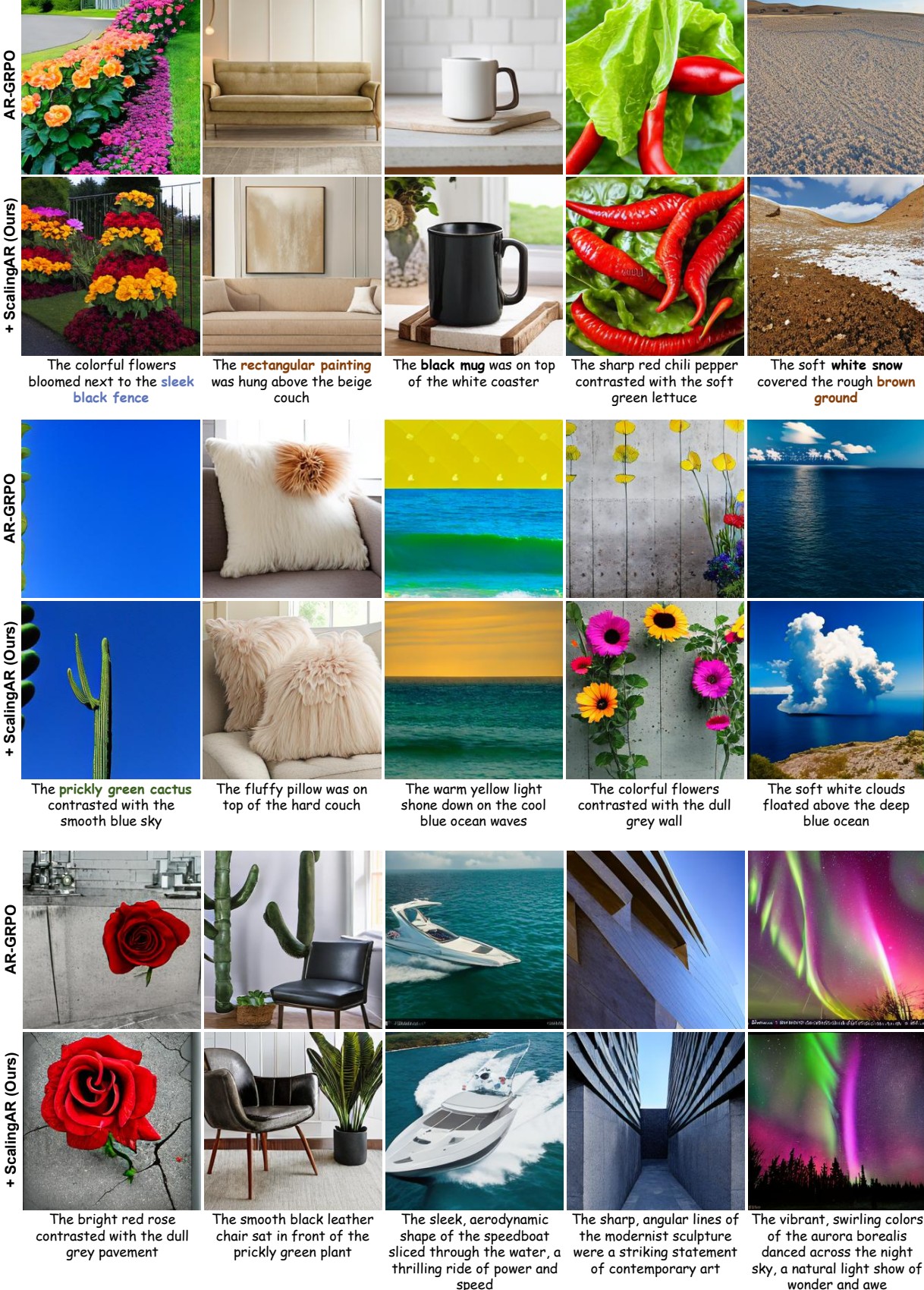

*Figure 12.* More results demonstrations of ScalingAR on AR-GRPO (Yuan et al., 2025).

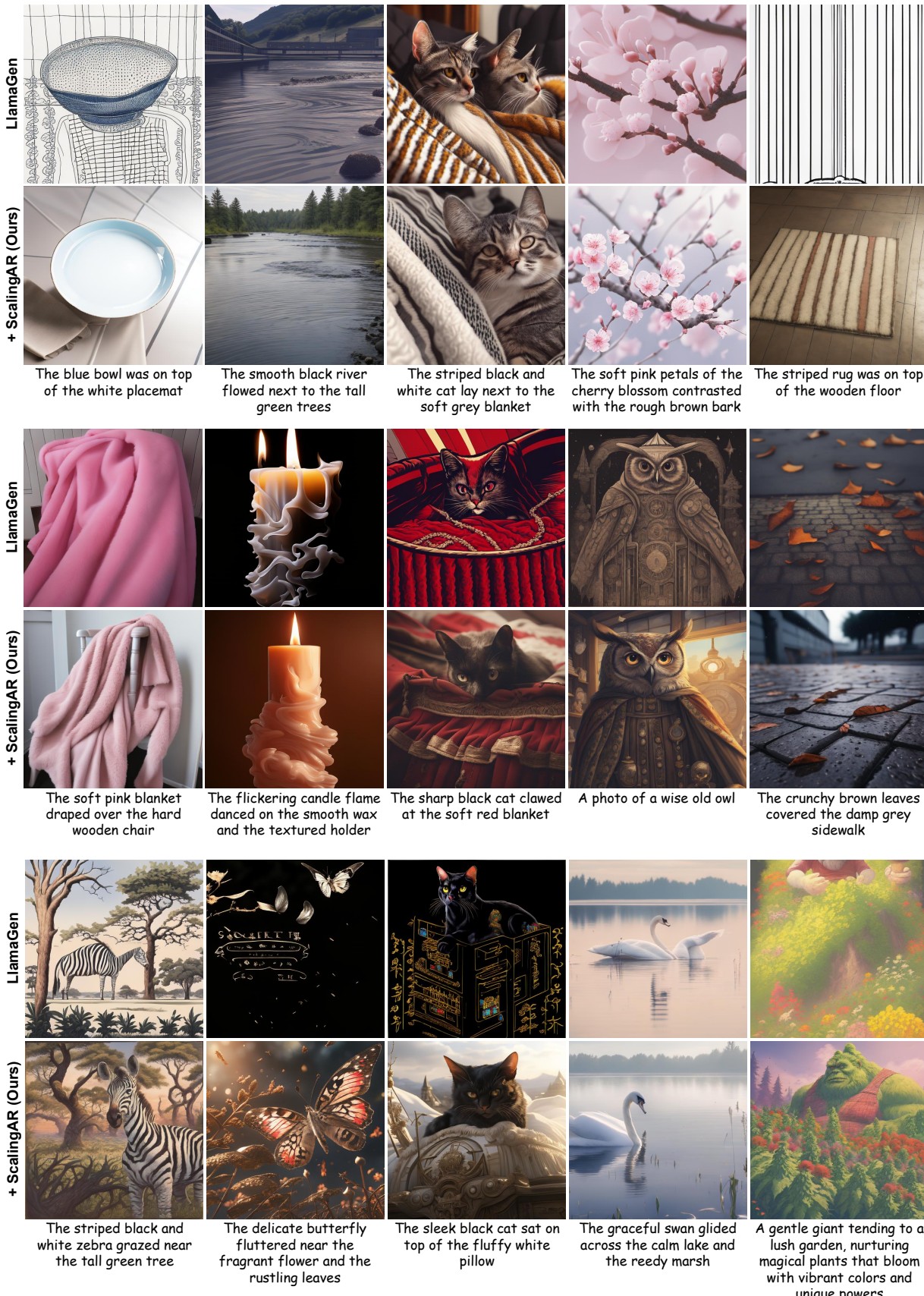

*Figure 13.* More results demonstrations of `ScalingAR` on LlamaGen (Sun et al., 2024).

