# OpenReview forum: "ScalingAR: Scaling Confidence for Autoregressive Image Generation"
_ICML.cc/2026/Conference — ICML 2026 regular_

### Official Review · Reviewer_8b5S · 2026-02-20

**Soundness:** 3
**Presentation:** 2
**Significance:** 3
**Originality:** 3
**Overall Recommendation:** 4
**Confidence:** 3

**Summary:**

The paper introduces ScalingAR, a novel test-time scaling (TTS) framework designed specifically for next-token prediction (NTP) autoregressive image generation models. Addressing the inefficiencies of traditional scaling methods like Best-of-N, which rely on external reward models and full image decoding, ScalingAR utilizes the model's intrinsic token entropy as a real-time confidence signal. The framework operates on two levels: a Profile Level that continuously monitors generation stability and semantic alignment (text prompt adherence), and a Policy Level that actively uses this data to dynamically adjust guidance strength and automatically prune unpromising generation trajectories. By intervening early rather than waiting for full images to generate, ScalingAR achieves significant performance gains, improving base models by 12.5% on GenEval and 15.2% on TIIF-Bench, while simultaneously reducing visual token consumption by 62.0% and enhancing robustness in complex scenarios.

**Compliance With Llm Reviewing Policy:**

Affirmed.

**Final Justification:**

Keeping score as the request of generalizing to the video modality is unresolved.

**Key Questions For Authors:**

Given the shared next-token prediction paradigm between recent autoregressive image models and emerging video architectures, I am curious if you have considered extending the ScalingAR framework to the temporal domain. The potential benefits seem substantial; specifically, applying your 'Adaptive Termination' policy could drastically cut the massive computational overhead inherent to video generation by pruning unpromising frame sequences early, while the 'Conditional Channel' could effectively combat the severe semantic drift and error accumulation that often plague long-form autoregressive generation. How would you approach adapting this spatial metric to evaluate 3D spatio-temporal blocks without falsely penalizing complex, high-frequency temporal dynamics, and what unique evaluation bottlenecks do you anticipate when scaling these intrinsic confidence signals across the significantly larger context windows required for temporally coherent video?

**Limitations:**

yes

**Strengths And Weaknesses:**

Strengths:
- ScalingAR substantially enhances both general and compositional image generation compared to base models. Specifically, it achieves a 12.5% performance improvement on the GenEval benchmark and a 15.2% improvement on the TIIF-Bench.
- Unlike traditional scaling methods (such as Best-of-N) that require generating multiple full images, ScalingAR is highly token-efficient. It successfully reduces visual token consumption by 62.0% while still outperforming baseline methods.
- The framework makes autoregressive models much more resilient when faced with difficult or complex instructions. In robustness tests, such as those using logically contradictory "impossible prompts," ScalingAR mitigated performance degradation by 26.0% compared to base models.

Weaknesses:
- The authors did not conduct experiments specifically to isolate how different batch sizes ($N$) affect the pruning mechanism. They evaluated different sample numbers $N = 1, 2, 4, 8$ only in the context of testing the overall scalability of the entire ScalingAR framework. As shown in Figure 5 (Right), they measured how total performance on the TIIF-Bench improved as $N$ increased, demonstrating that ScalingAR consistently outperforms baselines across these different widths. However, when they performed specific ablation studies dedicated to the Adaptive Termination (pruning) mechanism in Appendix D.2, they did not vary $N$. Instead, they kept the sample size fixed (their standard setup uses $N=8$) and only experimented with altering the internal pruning parameters. Because they only varied $N$ for the entire pipeline's performance rather than isolating the pruning gate, the paper lacks an empirical breakdown of how the quantile-based pruning mathematics function, or if they break down, when forced to operate on extremely small batches like $N=1$ or $N=2$.

---

> ### Author Rebuttal · Authors · 2026-03-30
>
> We sincerely thank you for your careful comments and thorough understanding of our paper! We are also encouraged that the reviewers recognized the clear motivation, the novel reward-free and lightweight design, and the strong performance, efficiency, and robustness of the proposed framework.
>
> Your insightful feedback has helped us clarify and refine our contributions. Here we give point-by-point responses to your comments:
>
> ---
> > `Weakness 1`: Robustness of quantile-based pruning at small batch sizes.
>
> We sincerely appreciate this insightful observation. You are correct that our previous ablations kept $N=8$ fixed, leaving the discrete mathematical behavior of quantile-based pruning at extreme low-width regimes ($N=1, 2$) under-explored. We will clarify this mechanism and add an isolated ablation in the revised Appendix.
>
> - **Behavior at small $N$:** The continuous pruning quantile $p$ translates into a discrete number of dropped trajectories via a floor function: $N_{\text{drop}} = \lfloor N \times p \rfloor$.
>
>     (1) **When $N=1$:** $N_{\text{drop}} = 0$, so comparative pruning is inactive and the system reduces to standard AR generation (with the guidance scheduler still active unless ablated).
>
>     (2) **When $N=2$:** depending on the exact $p$ and the hysteresis rule, the mechanism either keeps both trajectories or drops the weaker one, effectively reducing to one-trajectory selection when the confidence gap is sufficient.
>
> - **Isolated Ablation:** To empirically demonstrate this, we further conduct an ablation isolating the **Adaptive Termination Only** (without Guidance Scheduler) across varying $N$ on LlamaGen (GenEval).
>
>     *Table A: Isolated ablation of adaptive termination across varying $N$.*
>     |Setup|N|Token Consumption|Basic|Advanced|Designer|Overall|
>     |-|-|-|-|-|-|-|
>     |Base|1|2.8e+05|49.6|40.4|40.3|40.4|
>     |+ BoN|2|5.6e+05|**51.2**|40.6|38.4|41.2|
>     |**+ Termination Only (Ours)**|2|4.1e+05|50.7|**41.2**|**40.5**|**41.4**|
>     |+ BoN|4|1.1e+06|53.2|40.5|37.6|41.5|
>     |**+ Termination Only (Ours)**|4|5.0e+05|**53.8**|**42.0**|**41.6**|**43.0**|
>     |+ BoN|8|2.3e+06|54.8|40.8|37.7|42.0|
>     |**+ Termination Only (Ours)**|8|8.6e+05|**54.1**|**43.1**|**42.2**|**45.1**|
>
>     These results show that the pruning mechanism does not break at small $N$. Even at $N=2$, it reduces wasted computation relative to BoN while slightly improving overall performance. Its benefit becomes larger at $N \ge 4$, where the relative confidence distribution supports more effective pruning.
>
> ---
> > `Question`: Discussion on the potential of extending the framework to autoregressive video generation.
>
> We are deeply grateful for this exceptionally insightful and forward-looking question. We entirely agree that extending ScalingAR to the temporal domain is a highly promising frontier. You perfectly captured the potential benefits: cutting massive computational waste via Adaptive Termination and mitigating long-horizon error propagation via the Conditional Channel.
>
> Regarding your specific technical questions, we envision the following approaches:
>
> - **3D spatio-temporal profiling:** Complex, high-frequency temporal dynamics (*e.g.*, fast motion) naturally exhibit higher token entropy because they are harder to predict. To avoid falsely penalizing these valid dynamics, one possible direction is to track **temporal changes in entropy/confidence**, rather than relying on absolute entropy alone, so that smooth but complex motion is distinguished from abrupt structural degradation.
>
> - **Long-context evaluation:** Video generation uses much larger context windows, so confidence signals can become diluted over time. This suggests using **sliding-window** or **decayed** confidence accumulation so that recent local failures are not overwhelmed by earlier stable frames. In addition, maintaining multiple parallel trajectories in video would put much heavier pressure on memory and KV-cache efficiency, making early pruning even more important.
>
> We sincerely thank you for this inspiring discussion. We will incorporate these specific insights into the Future Work section of our revised manuscript. Once again, we sincerely thank you for this inspiring discussion and for the time and effort dedicated to evaluating our work.

---

> > ### Author Rebuttal · Reviewer_8b5S · 2026-04-03
> >
> > Keeping the score as I think pruning would be more effective when the generation is for a modality that has a time axis, for example: video.

---

> > > ### Author Response · Authors · 2026-04-03
> > >
> > > **Dear Reviewer 8b5S**,
> > >
> > > We sincerely appreciate your continued engagement with our work.
> > >
> > > Your intuition is absolutely spot-on: the computational savings and error mitigation of adaptive pruning will indeed be exponentially more impactful in temporal modalities like video. We view establishing this framework in the 2D spatial domain as a crucial foundational step toward realizing that potential, and we are thrilled that you share this exciting vision. **Your insights have heavily inspired our roadmap for future research**.
> > >
> > > Thank you once again for your invaluable time, expertise, and forward-looking feedback.
> > >
> > > Best regards,
> > >
> > > Authors

---

### Official Review · Reviewer_kyKf · 2026-03-09

**Soundness:** 2
**Presentation:** 2
**Significance:** 3
**Originality:** 3
**Overall Recommendation:** 2
**Confidence:** 3

**Summary:**

The paper focuses on the test time scaling methods for next token prediction autoregressive image generation models. The paper proposes Scaling AR, which is a test-time scaling method that leverages token-level entropy and conditional information to estimate confidence during the token generation stage. Based on this confidence signal, the method prunes the low-quality trajectories and adjusts the CFG during inference.

The main contributions are: 1) two-level confidence-based test time scaling using intrinsic model signals to improve generation quality without requiring any external reward model or early partial decoding. Experiments were performed on the GenEval TIIF-Bench dataset that show improvement over the considered baselines.

**Compliance With Llm Reviewing Policy:**

Affirmed.

**Final Justification:**

I still believe the paper's novelty is limited. The paper implies the LLM post-training strategies, which are not very tricky to implement. Hence, I still believe that the novelty part could be improved in this work. I intend to keep my rating.

**Key Questions For Authors:**

See weaknesses.

**Limitations:**

yes

**Strengths And Weaknesses:**

Strengths:

1. The paper proposes a new method for test time scaling using token entropy and conditional information to estimate token confidence during visual generation. The idea is well motivated by the instability of partial image decoding.

2. The lightweight confidence computation, when applied with adaptive trajectory pruning and dynamic guidance scheduling makes the overall process efficient and interpretable.

3. The method does not require any additional training or reward models, unlike many test-time scaling methods. This makes the method easily applicable to all types of AR image generation models.

Weaknesses:

1. **Limited Novelty:** Although the method has been developed for image generation models, several key ideas, such as entropy-based confidence estimation, have appeared in prior LLM inference and decoding strategies. The novelty, therefore, lies mainly in the adaptation to AR image models.

2. **Heuristic Nature:** Many design choices just appear with no proper theoretical motivation or empirical justification (EMAs, quantiles, normalizations etc). This makes the choices heuristic.

3. **Improper Baselines:** While the method only chooses one of the test time scaling methods from LLMs and applies it to visual autoregressive generation, it does not make any comments on other existing LLM test time scaling methods, why are they not applicable if they are applicable why they are not used as baselines. This makes the number of baselines very minimal (only IS and BoN as baselines).

4. The testing is performed only for autoregressive image generation models, why not for the autoregressive video generation models or multimodal models?

5. **Unclarity at multiple places:** The writing of the paper requires some serious rewriting and clarifications. For instance: 1) What is $E_k$ in equation 6, what is $\mathcal{N}_{\text{rolling}}$. 2) In equation 5, the EMA is over what is never written. 3) How to understand the Table 1, why EMU and Janus are not computed over TIIF-bench. 4) In Table 1 if the ScalingAR method is not applied over models shown in first 6 rows why are they even there?

---

> ### Author Rebuttal · Authors · 2026-03-30
>
> Thank you for your thoughtful and constructive feedback! We are encouraged that the reviewers recognized the motivation, design, and empirical strength of our framework.
>
> In response to your efforts, we have carefully prepared a point-by-point reply:
>
> ---
> > `Weakness 1`: Contextualizing novelty relative to LLM strategies.
>
> We appreciate your perspective. Token entropy is common in LLM inference, but applying it to NTP AR **image generation** is non-trivial because partial image prefixes provide weak early signals. Our contribution is not entropy alone, but a **probability-only test-time scaling** framework for AR image generation:
>
> - **Visual-specific profiling:** Unlike 1D text, images have 2D grid structures. We introduce *worst-block spatial stability* to capture localized visual anomalies, coupled with temporal tracking to filter transient noise.
>
> - **Conditional channel beyond entropy:** Entropy alone cannot detect semantic drift in image generation. We introduce a *Conditional Channel* that measures prompt adherence via KL-divergence between conditional and unconditional logits, a metric unique to guided visual generation.
>
> - **Active control:** These signals drive *adaptive trajectory pruning* (with recovery) and *dynamic CFG scheduling*, rather than being used only as passive decoding scores.
>
> We will clarify this distinction more explicitly in the revision.
>
> ---
> > `Weakness 2`: Motivation behind algorithmic components.
>
> We respectfully clarify that these choices are heuristic in form, but not arbitrary:
>
> - **EMA** filters step-level noise and stabilizes confidence estimates, reducing premature pruning.
>
> - **Quantile thresholding** uses relative ranking within the batch, avoiding brittle fixed cutoffs as confidence varies across prompts.
>
> - **Normalization** aligns signals with different scales (e.g., entropy and KL divergence) so they can be fused safely.
>
> We support these choices with ablations (`Table 2`), sensitivity analysis (`Appendix D.2`), and the **same default configuration** across **LlamaGen, AR-GRPO, SimpleAR, and Janus-Pro** (`Table 1` and `3`).
>
> ---
> > `Weakness 3`: Discussion on LLM test-time scaling.
>
> Most LLM test-time scaling methods are not directly applicable to NTP autoregressive **image generation**, so our baseline scope is constrained by **modality and interface mismatch**, not by omission.
>
> - **Reasoning-based** (*e.g.*, CoT) assume semantically meaningful intermediate tokens, while visual NTP tokens are spatial patches. While current prepending an external LLM to refine prompts is possible, this merely optimizes text conditioning rather than scaling the visual model's own inference capabilities.
>
> - **Search-based** (*e.g.*, MCTS, PRMs) require reliable scoring of intermediate states, but visual NTP intermediate states are spatially **incomplete** images with weak and unstable reward signals.
>
> We further verify this by adapting TTS-VAR, a recent MCTS-like scaling method, to our setting. Its need for frequent partial decoding causes much higher overhead and still worse performance than ScalingAR:
>
> *Table A: Comparison of TTS-VAR on GenEval with NVIDIA 140G H200 GPU.*
> ||N|Wall-clock (s)|Tokens/Img|Performance|
> |-|-|-|-|-|
> |LlamaGen|1|24.93|1024|40.4|
> |+ BoN|8|218.44|8192|42.0|
> |+ TTS-VAR|8|269.12|4562|41.6|
> |**+ ScalingAR (Ours)**|8|**69.56**|**2350**|**46.5**|
>
> This is why we use BoN and Importance Sampling as the **most representative paradigm-agnostic baselines** that can be applied to visual NTP without structural mismatch.
>
> ---
> > `Weakness 4`: Why not for the autoregressive video generation models or multimodal models?
>
> Extending ScalingAR to autoregressive **video** or **multimodal** generation is beyond the scope of this paper and is not a direct plug-and-play transfer.
>
> - **Video** requires spatio-temporal confidence profiling rather than the 2D spatial profiling used here.
>
> - **Multimodal models** involve heterogeneous token types and cross-modal interactions, so the same signals and policies may not transfer directly.
>
> Our focus here is to establish a **probability-only test-time scaling framework for AR image generation**, validated across four diverse base models. We will clarify this scope in the revision.
>
> ---
> > `Weakness 5`: Notations and table context.
>
> We appreciate you pointing out these details.
>
> - **Equations:** (1) **Eq.6:** $E_k$ is the entropy of the $k$-th spatial block; $H$ is the entropy function; $\mathcal{N}_\text{rolling}$ is a rolling min-max normalization. (2) **Eq.5:** The EMA is computed over generation steps to smooth step-wise sampling noise.
>
> - **Reference part of Table 1:** (1) **EMU/Janus missing TIIF-Bench:** EMU and Janus are shown only with established GenEval results for reference because no official TIIF-Bench pipeline is available. (2) **Purpose of the first 6 rows:** These are **reference models** included to contextualize the broader performance landscape; ScalingAR is applied only to the models in our main experimental setting.

---

> > ### Author Rebuttal · Reviewer_kyKf · 2026-04-02
> >
> > I still believe the paper's novelty is limited.

---

### Official Review · Reviewer_4CJW · 2026-03-10

**Soundness:** 2
**Presentation:** 3
**Significance:** 3
**Originality:** 2
**Overall Recommendation:** 4
**Confidence:** 4

**Summary:**

This study's key topic pertains to test-time scaling for autoregressive image generation. The paper proposes ScalingAR, a confidence-driven inference framework that uses token entropy to guide trajectory pruning and adaptive guidance during generation. Experiments on several autoregressive models and benchmarks show improvements in generation quality, efficiency, and robustness. This article's important contribution pertains to introducing an entropy-based test-time scaling strategy for autoregressive image generation without relying on external reward models.

**Compliance With Llm Reviewing Policy:**

Affirmed.

**Final Justification:**

The authors have satisfactorily addressed my concerns, and I will maintain my positive score.

**Key Questions For Authors:**

1. **Hyperparameter generalization across models.** The proposed framework introduces multiple hyperparameters and control policies. Can the same set of hyperparameters be consistently applied across different autoregressive models and datasets, or does the method require model-specific tuning?

2. **Comparison with Best-of-N sampling.** The paper shows that ScalingAR outperforms Best-of-N (BoN) sampling in the experiments. However, BoN theoretically explores a larger candidate space by generating multiple samples before selection. Could the authors provide more intuition or analysis explaining why BoN performs worse in this setting?

**Limitations:**

**No.** While the paper briefly discusses potential practical impacts (e.g., energy efficiency and the need for safety guardrails when deploying generative models), it does not sufficiently discuss the technical limitations of the proposed method. In particular, the paper could better address limitations such as the reliance on entropy-based confidence signals, the sensitivity to multiple hyperparameters, and the potential challenges of applying the method to larger models or different generation architectures.

**Strengths And Weaknesses:**

Strengths:

1. **Clear motivation and problem setting.** The paper addresses an important problem: how to design effective test-time scaling strategies for autoregressive image generation, which remains underexplored compared to language models.

2. **Reward-free test-time scaling.** Unlike many existing methods that rely on reward models or early decoding, the proposed approach uses intrinsic signals derived from token entropy, which simplifies deployment and reduces computational overhead.

3. **Good writting.** The paper is clearly written and well organized, making the proposed method and experimental results easy to understand.

Weaknesses:

1. The proposed framework is relatively complex and introduces many hyperparameters, which may make the method sensitive to tuning and potentially limit its robustness across different models and settings.

---

> ### Author Rebuttal · Authors · 2026-03-30
>
> We would like to express our deepest respect for your meticulous review! We are also encouraged that the reviewers recognized the clear motivation, the novel reward-free and lightweight design, and the strong performance, efficiency, and robustness of the proposed framework.
>
> Here we give point-by-point responses to your comments and describe the revisions we made to address them.
>
> ---
> > `Weakness & Question 1`: Hyperparameter generalization.
>
> Thank you very much for your thorough review! We agree that hyperparameter generalization is important for a test-time method. In our experiments, ScalingAR does **not** rely on model-specific tuning.
>
> - **Same configuration across models:** We use the **same default hyperparameters** mostly across four base models: LlamaGen (0.8B), AR-GRPO (0.8B), SimpleAR (1.5B), and Janus-Pro (1B). Despite differences in architecture and training objectives, ScalingAR consistently improves performance in `Table 1` and `3` **without** individual parameter adjustments.
>
> - **Stable structural priors:** Most listed settings are shared structural constants rather than heavily tuned knobs. The framework mainly relies on intrinsic signals (entropy and CFG-based KL divergence), and our sensitivity analysis in `Appendix D.2` shows small variation under reasonable changes.
>
> We will clarify this more explicitly in the revision.
>
> ---
> > `Question 2`: Intuition behind ScalingAR's superiority over Best-of-N.
>
> We acknowledge your insight. We respectfully clarify that the key difference is **how extra test-time compute is used**. BoN spends compute on generating multiple **complete** trajectories and selecting among them afterward, while ScalingAR uses compute **during generation** to preserve promising trajectories and **correct unstable** ones.
>
> - **Active rectification vs. passive selection**: BoN can only rank finished trajectories and cannot intervene once an error appears early in generation. In contrast, ScalingAR dynamically adjusts guidance through the **Guidance Scheduler** and prunes unstable branches through **Adaptive Termination**, which reduces error propagation during decoding.
>
> - **The Reward Model Bottleneck**: BoN is also limited by the quality and bias of the **external** selector. As shown in **Table B**, BoN with different reward models (CLIP, GPT, ImageReward) still underperforms ScalingAR, while ScalingAR relies on **intrinsic** confidence signals available during generation.
>
>     *Table B: Performance comparison between TTS strategies with different reward models.*
>     |Method|Reward Model|Basic|Advanced|Designer|Overall|
>     |-|-|-|-|-|-|
>     |LlamaGen|-|49.6|40.4|40.3|40.4|
>     |+ BoN|CLIP Score|52.0|40.4|36.9|40.8|
>     |+ BoN|GPT Score|53.2|41.0|40.7|43.2|
>     |+ BoN|ImageReward|54.8|40.8|37.7|42.0|
>     |**+ ScalingAR (w/o Scheduler)**|-|54.1|43.1|42.2|45.1|
>     |**+ ScalingAR (Ours)**|-|**57.4**|**44.1**|**42.5**|**46.5**|
>
>     As shown, ScalingAR consistently outperforms BoN across reward models. The additional gain from the Guidance Scheduler further shows that dynamic intervention during generation is more effective than post-hoc selection alone.
>
> ---
> > `Limitation`: Broadening the discussion on technical constraints.
>
> We agree that the technical limitations should be stated more clearly, and we will expand this discussion in the revision.
>
> - **Reliance on confidence proxies:** Entropy and CFG-based signals are effective proxies for instability, but they may not capture all semantic or perceptual failure modes.
>
> - **Scope of hyperparameter robustness:** Our current results show that *one default configuration* transfers well across *four* AR models, but this does not guarantee the same behavior for substantially different architectures.
>
> - **Architectural scope:** ScalingAR is currently designed for the NTP autoregressive setting. Extending the same idea to non-autoregressive models or other generation paradigms is non-trivial.
>
> These limitations will be discussed more explicitly in the revised manuscript.
>
> ---
> Thank you again for your valuable feedback. We hope this rebuttal resolves your concerns and enhances the clarity of our work. We remain open to any further suggestions for improvement.

---

> > ### Author Rebuttal · Reviewer_4CJW · 2026-04-04
> >
> > Thank you for the well-prepared rebuttal. The authors have satisfactorily addressed my concerns, and I will maintain my positive score.

---

> > > ### Author Response · Authors · 2026-04-04
> > >
> > > **Dear Reviewer 4CJW**,
> > >
> > > Thank you for confirming that our rebuttal resolved your concerns and for maintaining your positive score. We deeply appreciate your recognition of our **clear motivation and the practical value of our reward-free design**.
> > >
> > > Your constructive suggestions regarding hyperparameter generalization and technical limitations were invaluable. We will ensure these discussions are fully integrated into the revised manuscript.
> > >
> > > Thank you again for your time and meticulous review!
> > >
> > > Warm regards,
> > >
> > > Authors

---

### Official Review · Reviewer_gs5A · 2026-03-11

**Soundness:** 3
**Presentation:** 3
**Significance:** 3
**Originality:** 2
**Overall Recommendation:** 4
**Confidence:** 3

**Summary:**

This paper proposes ScalingAR, a test time scaling framework for next token prediction based autoregressive image generation. The main idea is to avoid external reward models and frequent early decoding by using only model internal signals to estimate generation confidence during sampling. The framework operates at two levels: (i) a Profile Level that integrates uncertainty and conditional signals into a unified confidence state, and (ii) a Policy Level that uses this state for adaptive trajectory pruning and dynamic guidance scheduling. Experiments demonstrate that ScalingAR significantly improves image quality on GenEval and TIIF-Bench while reducing visual token consumption.

**Compliance With Llm Reviewing Policy:**

Affirmed.

**Final Justification:**

The rebuttal addressed most of my concerns and improved my confidence in the paper. I am therefore raising my recommendation to Weak Accept.

**Key Questions For Authors:**

1. The ablation of the Policy Level is currently too lightweight for such a central contribution and should be moved to the main text. In particular, Adaptive Termination and Guidance Scheduler should be evaluated separately and jointly under matched token budgets, to disentangle algorithmic gains from pure compute reallocation.

2. Given the substantial wall-clock overhead introduced by parallel trajectories and the additional cost of unconditional forward passes for the conditional channel, it remains unclear whether the proposed dual-channel design significantly outperforms simpler reward-free intrinsic baselines. A more convincing comparison would evaluate ScalingAR against naive reward-free alternatives, such as ranking trajectories by cumulative log-likelihood or average confidence, or using a static entropy-based early termination rule, under strictly matched wall-clock budgets.

3. How sensitive is the framework's performance to the heuristic choices of spatial block size ($b$) and worst-block percentile ($q$), given their critical role in detecting localized failures? Stable behavior would improve confidence in the method.

**Limitations:**

yes

**Strengths And Weaknesses:**

**Strengths**

1.	The paper is well motivated by the challenge of early signal scarcity in NTP-based image generation, where unstable intermediate decoding makes test-time scaling based on early decoding or external reward signals difficult to apply effectively.

2.	The framework introduces a "zero-auxiliary" design that utilizes intrinsic signals (token entropy, spatial stability, and KL-divergence) to drive adaptive trajectory pruning and dynamic guidance scheduling.

3.	The paper shows improvements over baseline scaling methods across multiple base models, and also provides robustness analysis and ablation studies to support the effectiveness of the approach.

**Weaknesses**

1. The novelty claim appears overstated. In particular, the statement “We for the first time investigate token entropy in visual token generation” is difficult to support, since prior work such as ARsample [1] has already explored entropy in autoregressive image generation and used it to improve decoding quality and efficiency.  So the novelty here is the TTS control scheme for NTP image generation, not the first-ever use of entropy on visual tokens.

2. Although the paper discusses TTS-VAR, it does not include this line of work in the empirical comparison. Given its close relevance to visual autoregressive test-time scaling, a stronger justification for excluding it, or a discussion of whether a meaningful adaptation to the NTP setting is feasible, would make the comparison more convincing.

3.  ScalingAR depends on multiple hyperparameters and design choices, which are listed in the appendix, but the paper only analyzes a limited subset of them. This makes it difficult to assess how robust the method is to tuning.

4.  The framework combines several manually designed components, including dual channel fusion, quantile based thresholds, and recovery hysteresis, but provides limited theoretical justification for these choices. As a result, its stability and transferability across different model architectures or data distributions remain unclear.

[1] Towards better & faster autoregressive image generation: From the perspective of entropy. NeurIPS 2025.

---

> ### Author Rebuttal · Authors · 2026-03-30
>
> Thank you for your thoughtful and constructive feedback! We are encouraged that the reviewers recognized the motivation, design, and empirical strength of our framework.
>
> Here we give point-by-point responses to your comments:
>
> ---
> > `Weakness 1`: Clarification on the claim regarding visual entropy.
>
> We agree that the original “for the first time” wording was too strong. ARsample uses entropy for autoregressive *decoding*, while our work studies **test-time scaling** for NTP autoregressive image generation. In ScalingAR, entropy is *combined with conditional signals* for **trajectory pruning** and **dynamic guidance scheduling**, without partial decoding or external reward models. We will cite ARsample and clarify this distinction.
>
> ---
> > `Weakness 2`: Discussion on adapting TTS-VAR to the NTP paradigm.
>
> TTS-VAR is designed for **Next-Scale Prediction**, where intermediate states are spatially **complete** and can be scored by a reward model. In **Next-Token Prediction**, intermediate decodes are spatially incomplete, making reward-based filtering noisier. We therefore implement an **NTP-adapted TTS-VAR** baseline by pausing generation midway, decoding partial sequences with black-padded ungenerated regions, and applying ImageReward for filtering.
>
> *Table A: Comparison of TTS-VAR on TIIF-Bench with NVIDIA H200 GPU.*
> ||N|Wall-clock (s)|Tokens/Img|Performance|
> |-|-|-|-|-|
> |LlamaGen|1|24.93|1024|40.4|
> |+ BoN|8|218.44|8192|42.0|
> |+ TTS-VAR|8|269.12|4562|41.6|
> |**+ ScalingAR (Ours)**|8|**69.56**|**2350**|**46.5**|
>
> TTS-VAR is slower due to repeated partial decoding and reward evaluation, and it underperforms ScalingAR, suggesting limited transfer to NTP.
>
> ---
> > `Weakness 3 & 4`: Hyperparameters, design choices, and transferability.
>
> We agree that robustness to tuning and design choices is important.
>
> - **Design motivation, not arbitrariness:** The intrinsic and conditional channels capture complementary signals of **generation stability** and **prompt influence**; the quantile threshold and recovery mechanism make pruning **batch-adaptive**.
>
> - **Unified settings across models:** We do **not** retune the method for different models or datasets. The **same configuration** is used mostly across **LlamaGen, AR-GRPO, Janus-Pro, and SimpleAR**, including both structural constants (*e.g.*, block size, EMA rate) and the key parameters analyzed.
>
> - **Robustness and transferability:** The shared default configuration works across four models (`Tables 1` and `3`), with ablations for key modules and policies (`Table 2`; `Appendix D.2` / `Table 4`). The quantile-based threshold adapts to each batch instead of using a fixed cutoff.
>
> We will futher clarify this in the revision.
>
> ---
> > `Question 1`: Expanded policy level ablation under matched token budgets.
>
> We agree that the Policy Level is central and will move the detailed ablation (currently `Table 4`) to the main text. We also add a matched-budget ablation for Adaptive Termination, Guidance Scheduler, and their joint effect.
>
> *Table B: Matched-budget ablation on Policy Level.*
> ||N|Token Consum.|Bas.|Adv.|Des.|Overall|
> |-|-|-|-|-|-|-|
> |LlamaGen|1|2.8e+05|49.6|40.4|40.3|40.4|
> |+ BoN|3|8.4e+05|53.0|40.4|37.3|41.3|
> |+ BoN|8|2.3e+06|54.8|40.8|37.7|42.0|
> |+ Scheduler Only|8|2.3e+06|53.6|42.0|41.0|43.8|
> |+ Termination Only|8|8.6e+05|54.1|43.1|42.2|45.1|
> |**+ Joint(Ours)**|8|8.6e+05|**57.4**|**44.1**|**42.5**|**46.5**|
>
> At matched budgets, Termination Only outperforms BoN N=3 (**45.1 vs. 41.3**), Scheduler Only outperforms BoN N=8 (**43.8 vs. 42.0**), and the joint policy performs best at the lower budget.
>
> ---
> > `Question 2`: Comparison with simpler intrinsic baselines under matched wall-clock time.
>
> For CFG-based AR models, conditional and unconditional logits are **already** computed during inference, and our conditional-channel score is **derived from these outputs with negligible extra cost**. We also compare against simple reward-free baselines under a matched wall-clock budget on LlamaGen (ScalingAR: 69.56s).
>
> *Table C: Comparison under matched wall-clock budget.*
> ||N|GenEval|TIIF-Bench|
> |-|-|-|-|
> |Cumulative LL|3|0.33|40.4|
> |Static Entropy|8|0.32|41.5|
> |**ScalingAR (Ours)**|8|**0.36**|**46.5**|
>
> Under comparable wall-clock time, ScalingAR outperforms both baselines on both benchmarks, indicating that the gain comes from better trajectory scoring and adaptive pruning rather than extra compute alone.
>
> ---
> > `Question 3`: Sensitivity to spatial block size ($b$) and worst-block percentile ($q$).
>
> We observe only small variation across a reasonable range of both parameters, with fluctuations within **0.5** points on TIIF-Bench.
>
> *Table D: Sensitivity analysis of spatial priors ($b$ and $q$).*
> |Para.|Value|Performance|
> |-|-|-|
> |$b$|2|46.2|
> ||4 (Default)|46.5|
> ||8|46.0|
> |$q$|0.05|46.2|
> ||0.10 (Default)|46.5|
> ||0.20|46.1|
>
> This suggests that $b$ and $q$ are stable **structural priors** rather than sensitive tuning knobs, and we use the same settings across models.

---

> > ### Author Rebuttal · Reviewer_gs5A · 2026-04-02
> >
> > Thanks for the detailed reply. The rebuttal resolved my concerns with promising experimental results. I will raise the score for weak acceptance.

---

> > > ### Author Response · Authors · 2026-04-02
> > >
> > > **Dear Reviewer gs5A**,
> > >
> > > We sincerely appreciate your constructive evaluation of our work and your decision to raise the score. Your insightful suggestions regarding the matched-budget ablations and baseline comparisons were instrumental in helping us strengthen the empirical rigor of our framework. We are also deeply encouraged by your recognition of the **strong motivation behind addressing early signal scarcity and the value of our "zero-auxiliary" design**. It has been a privilege to engage with your feedback, which has significantly improved the final manuscript.
> > >
> > > Thank you once again for your time, expertise, and constructive review.
> > >
> > > Best regards,
> > >
> > > Authors

---

### Decision · Program_Chairs · 2026-04-30

**Decision:**

Accept (regular)

**Comment:**

After the rebuttal period, one reviewer upgraded from Weak Reject to Weak Accept, resulting in three weak accepts and one reject. Reviewers agreed that the paper addresses a timely problem in autoregressive image generation and that the reward-free test-time scaling design is well motivated. The strongest positive feedback highlighted the consistent gains on GenEval and TIIF-Bench, the reduction in token consumption, and the practical value of using intrinsic confidence signals instead of external reward models. The rebuttal substantively strengthened the empirical case by adding an adapted TTS-VAR comparison, matched-budget ablations for the policy components, comparisons against simpler intrinsic baselines, sensitivity analyses for the pruning hyperparameters, and additional clarification on the small-batch behavior of the termination rule. However, one reviewer continued to argue that the conceptual novelty is limited, viewing the method as a relatively heuristic adaptation of existing entropy-based or LLM-style post-training and decoding strategies. Some residual concerns also remain about the number of control knobs in the framework and the scope of evaluation beyond autoregressive image generation, especially toward video or multimodal settings. On balance, the rebuttal resolved the main experimental concerns for the supportive reviewers, while the remaining objections are primarily about novelty and scope rather than empirical soundness. The AC recommends acceptance.